# A Single-Swap Local Search Algorithm for $k$-means of Lines

**Ting Liang**[1]**, Xiaoliang Wu**[1]**, Junyu Huang**[1]**, Jianxin Wang**[1,2]**, Qilong Feng**[1,*]

[1]School of Computer Science and Engineering, Central South University
[2]The Hunan Provincial Key Lab of Bioinformatics, Central South University,
Changsha 410083, China
tingliang@csu.edu.cn, xiaoliangwucsu@163.com, junyuhuangcsu@foxmail.com,
jxwang@mail.csu.edu.cn, csufeng@mail.csu.edu.cn

## Abstract

Clustering is a fundamental problem that has been extensively studied over past few decades, with most research focusing on point-based clustering such as $k$-means, $k$-median, and $k$-center. However, numerous real-world applications, such as motion analysis, computer vision, and missing data analysis, require clustering over structured data, including lines, time series and affine subspaces (flats), where traditional point-based clustering algorithms often fall short. In this paper, we study the $k$-means of lines problem, where the input is a set $L$ of lines in $\mathbb{R}^d$, and the goal is to find $k$ centers $C$ in $\mathbb{R}^d$ such that the sum of squared distances from each line in $L$ to its nearest center in $C$ is minimized. The local search algorithm is a well-established strategy for point-based $k$-means clustering, known for its efficiency and provable approximation guarantees. However, extending local search algorithm to the $k$-means of lines problem is nontrivial, as the capture relation used in point-based clustering does not generalize to the line setting. This is because that the point-to-line distance function lack the triangle inequality property that supports geometric analysis in point-based clustering. Moreover, since lines extend infinitely in space, it is difficult to identify effective swap points that can significantly reduce the clustering cost. To overcome above obstacles, we introduce a *proportional capture relation* that links optimal and current centers based the assignment proportions of lines, enabling a refined analysis that bypasses the triangle inequality barrier. We also introduce a *CrossLine* structure, which provides a principled discretization of the geometric space around line pairs, and ensures coverage of high-quality swap points essential for local search, thereby enabling effective execution of the local search process. Consequently, based on the proposed components, we develop the first single-swap local search algorithm for the $k$-means of lines problem, achieving a $(500 + \varepsilon)$-approximation in polynomial time for low-dimensional Euclidean space.

## 1 Introduction

Clustering is one of the most popular problems in machine learning, and has lots of applications in data mining, image classification, etc. The goal of clustering is to partition a given set of points into several disjoint clusters such that similar points end up in the same cluster, and dissimilar points are separated into different clusters [23]. Among classical clustering models [11, 12, 25], $k$-means is one of the most extensively studied, aiming to minimize the total squared distance from each data point to

---

*Corresponding Author

39th Conference on Neural Information Processing Systems (NeurIPS 2025).

its nearest center. More formally, given a set $P \subseteq \mathbb{R}^d$ of $n$ data points in a $d$-dimensional Euclidean space and a positive integer $k$, the goal of $k$-means is to find a set $C \subset \mathbb{R}^d$ of $k$ centers such that the objective $\sum_{p \in P} \min_{c \in C} \|c - p\|^2$ is minimized.

Although the $k$-means problem is NP-hard [21, 15], numerous approximation algorithms have been developed. One of the most widely used heuristics in practice is Lloyd's algorithm [17], even though it lacks provable worst-case approximation guarantees under general data distributions. To establish approximation guarantees, researchers introduced a series of algorithms based on primal-dual and randomized rounding techniques [2, 10], among which a primal-dual method leveraging nested quasi-independent sets [5] achieves the best-known approximation ratio of 5.912. Meanwhile, local search has emerged as a practical and theoretically grounded framework for the $k$-means problem. Several local search algorithms [6, 7] achieved a $(1 + \varepsilon)$-approximation under the assumption of fixed dimension $d$ or a constant number of clusters $k$. To further enhance the initialization, Arthur and Vassilvitskii [3] proposed an efficient seeding algorithm, known as $k$-means++, which achieves an $O(\log k)$-approximation in expectation. Subsequent studies [26, 1, 19] showed that $k$-means++ achieves constant-factor approximations when allowed to open $O(k)$ centers. Building upon this, Lattanzi and Sohler [16] proposed the LS++ algorithm by combining $k$-means++ seeding with local search, achieving a constant-factor approximation in $O(ndk^2 \log \log k)$ time. Choo *et al.* [4] further improved its efficiency, demonstrating that an $O(1/\varepsilon^3)$-approximation can be obtained using only $O(\varepsilon k)$ local search iterations. Under the assumption that each optimal cluster has size $\Omega(n/k)$, Huang *et al.* [13] proposed a fast and practical local search algorithm for $k$-means problem, and achieved a $(100 + \varepsilon)$ approximation in expectation. More recently, Huang *et al.* [14] gave the first multi-swap local search algorithm with running time linearly dependent on the data size, and achieved a $\left(50\left(1 + \frac{1}{t}\right) + \varepsilon\right)$-approximation with any swap size $t \geq 2$ in time $O\left(ndk^{2t+1} \log\left(\varepsilon^{-1} \log k\right)\right)$, improving the previous 509-approximation under linear-time constraints.

While these results provide strong theoretical and practical foundations for point-based clustering, many real-world applications involve structured data that are more naturally represented as geometric objects such as one-dimensional affine subspaces (i.e., lines). For example, in computer vision [18, 20], when multiple cameras are used to observe a set of fixed objects, each camera provides a directional observation, that is, a direction vector from the camera to the object. By combining each camera with its corresponding direction, a set of lines can be obtained. Clustering these lines helps identify different objects and recover their locations, since the lines corresponding to the same object tend to intersect at or near the object's true location. This motivates the study of clustering problems where the input consists of lines rather than points. Although conceptually similar to classical point-based formulations, line clustering introduces unique algorithmic and geometric challenges, and remains relatively underexplored in the literature. Among the few existing studies, Har-Peled and Varadarajan [8] studied the problem of finding the minimum intersection radius of a set of lines or affine subspaces, and introduced a substitute for the triangle inequality based on a novel analog of Helly's theorem. Aronov *et al.* [9] studied the problem of finding $k$ minimum-radius balls in Euclidean space that intersect all given lines, and developed a 2-approximation algorithm for $k = 2, 3$ with quasi-linear running time. Ommer *et al.* [22] considered the $k$-means of lines problem, and proposed the first heuristic algorithm without theoretical guarantee and time constraints. Perets *et al.* [24] proposed an approximation algorithm in $\mathbb{R}^2$ that starts with a bi-criteria approximate solution, and then uses the coresets technique, achieving a $(1 + \varepsilon)$-approximation in time $O(n(\log n/\varepsilon))^{O(k)}$ for any given parameter $\varepsilon > 0$. Further, Marom *et al.* [20] developed an algorithm for the $k$-means of lines problem that computes an $\varepsilon$-coreset of size $O(dk^{O(k)} \log(n)/\varepsilon^2)$ in time $O(d^3 n \log(n)k \log k + (d/\varepsilon)^2 + ndk^{O(k)})$.

In this paper, we focus on the $k$-means of lines problem. Given a set $L$ of $n$ lines in $\mathbb{R}^d$ and an integer $k \in \mathbb{N}_{\geq 1}$, the objective is to find a set of $k$ centers $C \subset \mathbb{R}^d$ that minimizes the total squared distance from each line $\ell \in L$ to its nearest center in $C$. In this paper, we present the first local search algorithm with provable approximation guarantees for the line-based setting. Although local search has been extensively analyzed for point-based $k$-means due to its simplicity and strong empirical performance, extending it to the line setting introduces several fundamental challenges. To motivate our algorithm, we first review the classical local search framework and highlight the key obstacles in adapting it to line clustering.

- The standard analysis of local search algorithm for $k$-means mainly relies on a capture relation, in which each optimal center is captured by the nearest center in the current

solution, such that it is easy to analyze the change in clustering cost incurred when points are reassigned in the process of local search. However, the above relation is not workable for the line setting, since the assignments of lines may be determined by an optimal center that is not the closest in Euclidean distance, and reassigning lines based solely on geometric closeness can lead to a significant increase in clustering cost.

- The sampling-based method is a commonly used technique in local search. However, unlike the point setting, it is more challenging to sample a point close to the optimal center from these lines, since the optimal center is unknown and each line spans a continuous space of candidate points, even we show that the cost of a high-cost cluster can be effectively approximated by lines that are close to their optimal center in the $k$-means of lines problem.

**Our Contributions.** To overcome these obstacles, we introduce a new proportional capture relation and a structured sampling technique tailored to the geometry of lines. Specifically, we define that an optimal center is captured by a current center if it receives the largest fraction of lines from that center, enabling a more faithful representation of line-to-center influence. To facilitate effective sampling, we further design a discretization structure, called *CrossLine*, which guarantees the existence of a point near the optimal center within a bounded region defined by two sampled lines. This structure allows us to recover, with provable guarantees, a good replacement center from a finite set of representative points. We formalize the proposed *proportional capture relation* and *CrossLine* structure in Section 3, and discuss some key properties. Building on these foundations, we develop the first local search algorithm for the $k$-means of lines problem with provable approximation guarantees. The main contributions of this paper are summarized as follows:

- In two-dimensional space, we introduce a *proportional capture relation* and a geometry-aware discretization structure *CrossLine*, both tailored to the challenges of line-based clustering. These tools enable an accurate representation of line-to-center assignments, and support efficient sampling from continuous line domains.

- In high-dimensional space, the existence of skew lines makes the *CrossLine* structure in $\mathbb{R}^2$ inadequate. To address this issue, we redefine the *CrossLine* structure to accommodate high-dimensional geometry, ensuring that the coverage guarantees remain valid under arbitrary line orientations.

- Building upon these tools, we propose the first single-swap local search algorithm for the $k$-means of lines problem, achieving a $(500 + \varepsilon)$-approximation, and running in polynomial time on low-dimensional Euclidean space.

- Extensive experiments on both synthetic and real-world datasets show that our algorithm consistently outperforms coreset-based baselines in both efficiency and clustering quality.

Formally, we have the following results of this paper.

**Theorem 1.** *For any $\varepsilon > 0$, there exists a local search algorithm for the $k$-means of lines problem, which achieves a $(500 + \varepsilon)$-approximation in expectation with polynomial time on low-dimensional Euclidean space.*

## 2 Preliminaries

For any positive integer $m \in \mathbb{N}^{\geq 1}$, let $[m] = \{1, \ldots, m\}$. Given a set $P \subseteq \mathbb{R}^d$ of points, for any $p, q \in P$, let $\delta(p, q) = \|p - q\|_2^2$ denote the square Euclidean distance between $p$ and $q$. Given any two set $A, B \subseteq \mathbb{R}^d$ of points, for any $p \in A$, let $\delta(p, B) = \min_{q \in B} \delta(p, q)$ denote the shortest square distance from $p$ to any point in $B$. Further, let $\delta(A, B) = \min_{p \in A, q \in B} \delta(p, q)$ denote the square distance between $A$ and $B$, and let $\Delta(A, B) = \sum_{p \in A} \delta(p, B)$ denote the sum of distances from each point in $A$ to its nearest point in $B$. Given a nonempty subset $C \subseteq P$ of centers, for any $c \in C$ and a radius $r$, a ball $\mathcal{B}(c, r)$ is the set of points that are within a distance $r$ from $c$, i.e., $\mathcal{B}(c, r) = \{p \in P \mid \delta(c, p) \leq r\}$. Given a set $L$ of lines in $\mathbb{R}^d$ and a set $C \subseteq \mathbb{R}^d$ of points, for any $\ell \in L$, let $\delta(\ell, C) = \min_{c \in C} \delta(\ell, c)$, where $\delta(\ell, c)$ is the shortest square distance from any point on the line $\ell$ to $c$. Formally, the $k$-means of lines (denoted as $k$-ML) problem considered in this paper can be defined as follows.

---
**Algorithm 1:** SLS-$k$-ML
---
**Input:** An instance $(L, d, k)$ of the $k$-ML problem and a parameter $T$
**Output:** A set $C \subset \mathbb{R}^d$ of at most $k$ centers
**1** $P \leftarrow$ CENTROID-SET$(L)$;
**2** $C \leftarrow$ Sample $k$ points from $P$ randomly;
**3 for** $i \in \{1, 2, \ldots, T\}$ **do**
**4** $\quad$ Sample two lines $\ell_1, \ell_2$ from $L$ with probability $b_\ell = \frac{\Delta(\{\ell\}, C)}{\sum_{\ell' \in L} \Delta(\{\ell'\}, C)}$ ($\ell = \{\ell_1, \ell_2\}$);
**5** $\quad$ $\mathcal{M} \leftarrow$ the point set returned by *CrossLine* of $(\ell_1, \ell_2)$;
**6** $\quad$ **for** each point $p \in \mathcal{M}$ **do**
**7** $\quad\quad$ **if** $\exists\, q \in C$, *s.t.* $\Delta(L, C \backslash \{q\} \cup \{p\}) < \Delta(L, C)$ **then**
**8** $\quad\quad\quad$ $C \leftarrow C \backslash \{q\} \cup \{p\}$;

**9 return** $C$.

---

---
**Algorithm 2:** CENTROID-SET
---
**Input:** A finite set $L$ of $n$ lines in $\mathbb{R}^d$
**Output:** A set $P$ of points
**1** $P \leftarrow \emptyset$;
**2 for** each line $\ell \in L$ **do**
**3** $\quad$ $P_\ell \leftarrow \emptyset$;
**4** $\quad$ **for** each line $\ell' \in L \backslash \{\ell\}$ **do**
**5** $\quad\quad$ $\pi_{\ell'}(\ell) \leftarrow$ the closest point on $\ell$ to $\ell'$;
**6** $\quad\quad$ $P_\ell \leftarrow P_\ell \cup \{\pi_{\ell'}(\ell)\}$;
**7** $\quad$ $P \leftarrow P \cup P_\ell$;

**8 return** $P$.

---

**Definition 1** (the $k$-ML problem). *Given a set $L \subseteq \mathbb{R}^d$ of $n$ lines in a $d$-dimensional Euclidean space and a positive integer $k$, the goal is to find a set $C \subset \mathbb{R}^d$ of $k$ centers such that the objective $\Delta(L, C) = \sum_{\ell \in L} \delta(\ell, C)$ is minimized.*

In this paper, we assume that all lines in $L$ are pairwise non-parallel (possibly intersecting or skew), ensuring distinct directional relationships and avoiding degenerate configurations. Given an instance $(L, d, k)$ of the $k$-ML problem, $C$ is called a feasible solution of this instance if $C \subseteq \mathbb{R}^d$ is a set of $k$ points. Throughout this paper, let $C^* = \{c_1^*, \ldots, c_k^*\}$ be the optimal solution and $\{L(c_1^*), \ldots, L(c_k^*)\}$ be the corresponding optimal clusters by assigning each line in $L$ to its closest centers in $C^*$. Let $\tau = \Delta(L, C^*)$ be the cost of the optimal solution $C^*$.

## 3 Local Search Algorithm with Single-Swap Strategy

The local search algorithm for the $k$-means problem typically begins with a candidate set of centers and iteratively replace one or more centers, as in single-swap or multi-swap strategies, to progressively reduce the clustering cost. Unlike classical $k$-means problem, which benefits from effective initialization algorithms such as $k$-means++ [16], the $k$-ML problem lacks a well-established algorithm for constructing a high-quality initial solution. Therefore, it is necessary to explicitly construct an initial solution for the $k$-ML problem. Moreover, extending local search algorithm to the $k$-ML problem introduces additional challenges beyond initialization. Specifically, the capture relation used in $k$-means problem does not hold in the line setting, as line assignments are governed by projection distances, and reassigning a line to the optimal center closest to its current center may lead to an increase in clustering cost. In addition, it is more challenging to sample a point that is close to an optimal center from given lines, since the optimal center is unknown and each line spans a continuous space of candidate points.

To overcome these challenges, we develop a single-swap local search algorithm for the $k$-ML problem, referred to SLS-$k$-ML, which is presented in Algorithm 1. Our algorithm SLS-$k$-ML consists of

two stages. In the first stage (steps 1–2), the algorithm starts with a centroid set $P$ obtained by CENTROID-SET that reduces the lines in $L$ to a set of points. Then, we obtain a feasible solution $C$ from the constructed centroid set $P$ with an approximation guarantee. In the second stage (steps 3-8), we execute the single-swap local search with $T$ iterations. In each iteration, the algorithm first to sample two lines $\ell_1, \ell_2$ from $L$ with a probability, and then constructs candidate swap pairs based on the *CrossLine* structure with respect to $\ell_1, \ell_2$ to find improvements. The *CrossLine* structure ensures that each optimal center can be effectively approximated by at least one grid point. Under the proposed *proportional capture relation*, such a grid point can be used to replace a current center, resulting in a reduction of the clustering cost.

## 3.1 Construct an Initial Solution

This section shows how to construct an initial solution with an approximation guarantee for a given instance of the $k$-ML problem in polynomial time. It is known that the local search algorithm typically begins with an initial solution. However, the $k$-ML problem lacks a well-established algorithm for generating a high-quality initial solution. Therefore, we first need to construct such an initial solution as a foundation for the local search process. To this end, we first construct a candidate point set $P$ by the CENTROID-SET procedure from [20] (see Algorithm 2). Given a set $L$ of $n$ lines in $\mathbb{R}^d$, for each line, CENTROID-SET computes its the closest points to all other lines, and collects them into a candidate set $P$ of $O(n^2)$ points. We then obtain the initial solution $C$ by sampling $k$ points uniformly at random from $P$. In the following, we show that the initial solution $C$ provides a constant-factor approximation to the optimal clustering cost, and can be computed in polynomial time.

**Lemma 2.** *Given an instance $\mathcal{I} = (L, d, k)$ of the $k$-ML problem, let $C$ be the set of points returned by step 2 of* SLS-$k$-ML. *Then, for a finite constant $\rho$, we have $\Delta(L, C) \leq \rho \cdot \Delta(L, C^*)$ (see Appendix A.1 for proof).*

## 3.2 Single-Swap Local Search in $\mathbb{R}^2$

To facilitate understanding of our proposed algorithm, this section focuses on the single-swap local search process in $\mathbb{R}^2$. Specifically, we construct a swap pair between the center in $C$ and a candidate center in $\mathcal{M}$ obtained by the *CrossLine* structure, which is related to two lines sampled from $L$ with a specified probability. If the clustering cost is reduced, we update the current solution by replacing the center in $C$ with the candidate center from $\mathcal{M}$. In the following, we show that if the current solution has a high cost (greater than $500\tau$), then the clustering cost can be reduced with a certain probability in each iteration.

**Lemma 3.** *Let $C \subseteq \mathbb{R}^2$ be a set of centers with $\Delta(L, C) > 500\tau$, and let $C'$ be the set of centers obtained in the $t$-th $(t \in [T])$ iteration of* SLS-$k$-ML *in step 8. Then, with probability at least $\Omega(\zeta^{-1})$, we have $\Delta(L, C') \leq (1 - \frac{1}{100k})\Delta(L, C)$, where $\zeta$ is a constant.*

Before proving Lemma 3, we first analyze a single-swap (w.l.o.g. $t$-th iteration) in steps 3–8 of SLS-$k$-ML. Let $C = \{c_1, \ldots, c_k\}$ denote the set of centers before the $t$-th swap operation, and let $\{L(c_1), \ldots, L(c_k)\}$ be the corresponding partition of $L$ induced by assigning the lines to the closest center in $C$. For clarity, when the specific index is not relevant, we omit it and simply write, for example, $c \in C$. Motivated by the local search for $k$-means [16], we adopt a similar strategy to analyze the cost reduction by comparing the current and optimal center sets. Specifically, the algorithm in [16] introduces the notion of a capture relation, which states that an optimal center can be captured by the nearest center in the current solution. However, the above relation relies on a point-to-point assignment setting, and is not workable for our problem, since the data consist of lines and the clustering cost is based on projection distances. For example, in the $k$-means setting, for each $c^* \in C^*$, assume that $c$ is the closest center in $C$ to $c^*$. By the capture relation, the optimal center $c^*$ is captured by $c$, which means that $c$ is the nearest center to $c^*$ among all centers in $C$. In the process of analyzing local search, we consider replacing $c$ with a point near to $c^*$, and reassigning the points in the cluster of $c$, thereby achieving a reduction in clustering cost. However, in the line setting, the lines in the cluster of $c$ may be significantly closer to another optimal center in $C^*$. In such case, if we reassign these lines to a point near to $c^*$, the clustering cost will drastically increase, indicating that the original capture relation does not generalize to the $k$-ML problem. Therefore, we propose a *proportional capture relation* better suited to the line setting: an optimal center $c^* \in C^*$ is captured by a center $c \in C$ if $c$ is assigned the largest proportion of lines from the cluster of $c^*$ among all center in $C$. Clearly, a center $c \in C$ may capture more than one optimal centers, and every optimal

center is captured by exactly one center from $C$. For a center $c \in C$, let $\mathcal{N}(c)$ be the set of centers in $C^*$ captured by $c$. For each optimal center $c^* \in C^*$, let $\mathcal{N}^{-1}(c^*)$ be the center in $C$ capturing $c^*$. Further, we introduce two types of matched swap pairs that correspond to the two cases encountered in local search as follows.

**The type-1 matched swap pair:** For any $c \in C$, if $|\mathcal{N}(c)| = 1$ (i.e., $\mathcal{N}(c)$ contains only one optimal center $c^*$), we define $(c, c^*)$ as a type-1 matched swap pair. Let $C_S$ denote all centers in $C$ satisfying the property of type-1 matched swap pair. For simplicity, we introduce a mapping $\psi_S : C_S \to C^*$ that maps each center in $C_S$ to an optimal center in $C^*$ such that $(c, \psi_S(c))$ is a type-1 matched swap pair. For each type-1 matched swap pair $(c, \psi_S(c))$ $(c \in C_S)$, we replace the current center $c$ with a point near the optimal center $\psi_S(c)$. For a type-1 matched swap pair, the lines in cluster $c$ $(c \in C_S)$ can be divided into two groups: The first group consists of lines that belong to both the current cluster of $c$ and the optimal cluster of $\psi_S(c)$. These lines are reassigned to a point near their optimal center, leading to a reduction in clustering cost. The second group consists of lines that belong to in the current cluster of $c$ but not to the optimal cluster of $\psi_S(c)$. These lines are reassigned to other centers in $C$, which may increase the clustering cost.

**The type-2 matched swap pair:** For any $c \in C$, if $|\mathcal{N}(c)| = 0$ (i.e., $c$ does not capture any optimal center), we can find another center $c' \in C$ with $|\mathcal{N}(c')| > 1$, and assign one of its captured optimal center $c^* \in \mathcal{N}(c')$ to form a type-2 matched pair $(c, c^*)$. Let $C_N$ denote all centers in $C$ satisfying the property of type-2 matched swap pair. For simplicity, we introduce a mapping $\psi_N : C_N \to C^*$ that maps each center in $C_N$ to an optimal center in $C^*$ such that $(c, \psi_N(c))$ is a type-2 matched swap pair. For a type-2 matched swap pair $(c, \psi_N(c))$ $(c \in C_N)$, we replace the lonely center $c$ with a point near the optimal center $\psi_N(c)$. Similar to the type-1 case, the lines in cluster $c$ $(c \in C_N)$ also can be divided into two groups: The first group consists of lines that belong to the optimal cluster of $\psi_N(c)$. These lines are reassigned to a point near their optimal center, leading to a reduction in clustering cost. The second group consists of lines that belong to the current cluster of $c$. These lines are reassigned to other centers in $C$, which may increase the clustering cost.

Through this construction, each optimal center in $C^*$ is paired with a current center in $C$ via exactly one type-1 or type-2 matched swap pair. Each swap pair involves both clustering cost reduction (from lines reassigned to a sampled point near their optimal center) and clustering cost increase (from lines reassigned to potentially farther centers). In the following, we will argue that with high probability, a point can be sampled to perform the swap operation such that the resulting reduction in clustering cost significantly bigger than the corresponding increase. For a type-1 or type-2 matched swap pair, we refer to the clustering cost increase as the reassignment cost, which is formally defined below.

**Definition 2.** *For any $c \in C_S$, the reassignment cost of the type-1 matched swap pair $(c, \psi_S(c))$ is defined as $\eta_1(c) = \Delta(L\backslash L(c^*), C\backslash\{c\}) - \Delta(L\backslash L(c^*), C)$, where $c^* = \psi_S(c)$. Similarly, for any $c \in C_N$, the reassignment cost of the type-2 matched swap pair $(c, \psi_N(c))$ is defined as $\eta_2(c) = \Delta(L, C\backslash\{c\}) - \Delta(L, C)$.*

**Lemma 4.** *For any $c \in C_S$ (or $c \in C_N$), the reassignment cost $\eta_1(c)$ (or $\eta_2(c)$) is at most $\frac{1}{5}\Delta(L(c), C) + 24\Delta(L(c), C^*)$ (see Appendix A.2 for proof).*

Lemma 4 implies that there exists a good bound on the reassignment cost. To proceed, we discuss the following two cases based on the cost contributions of type-1 and type-2 matched pairs: **Case 1:** $\sum_{c \in C_S} \Delta(L(\psi_S(c)), C) > \frac{1}{3} \cdot \Delta(L, C)$. **Case 2:** $\sum_{c \in C_S} \Delta(L(\psi_S(c)), C) \le \frac{1}{3} \cdot \Delta(L, C)$, thus $\sum_{c \in C^*\backslash\cup_{c \in C_S}\psi_S(c)} \Delta(L(c), C) \ge \frac{2}{3} \cdot \Delta(L, C)$.

### 3.2.1 The Analysis of Case 1

Recall that the definition of type-1 matched swap pair $(c, \psi_S(c))$ $(c \in C_S)$, we replace the current center $c$ with a point near the optimal center $\psi_S(c)$, with the goal of achieving a significant reduction in clustering cost. We refer to such clusters that yield a significant cost reduction as *good clusters*, and we formalize this notion below.

**Definition 3** (*Good type-1 cluster*). *For any $c \in C_S$, the cluster $L(c)$ is called a good type-1 cluster, if $\Delta(L(\psi_S(c)), C) - \eta_1(c) - 9\Delta(L(\psi_S(c)), \{\psi_S(c)\}) > \frac{1}{100k}\Delta(L, C)$. Otherwise, $L(c)$ is called a bad type-1 cluster with $(c, \psi_S(c))$.*

For simplicity, let $C_S^g$ denote the set of centers in $C_S$ for which the corresponding cluster is a good type-1 cluster. Let $C_S^b = C_S\backslash C_S^g$. The Definition 3 estimates the cost reduction of replacing $c$ with a

point close to $\psi_S(c)$ by considering a cluster that reassigns the lines in $L(c) \backslash L(\psi_S(c))$, and assigns all lines in $L(\psi_S(c))$ to the new center. We now aim to show that a good type-1 cluster can be sampled with high probability. To this end, we first argue that the total clustering cost contributed by good type-1 clusters is sufficiently large. Then, we show that with good probability, we can sample a point that is close to the center.

**Lemma 5.** *If* $3 \sum_{c \in C_S} \Delta(L(\psi_S(c)), C) > \Delta(L, C)$, *then we have* $\sum_{c \in C_S \cap C_S^g} \Delta(L(\psi_S(c)), C) \geq \frac{1}{50} \cdot \Delta(L, C)$ *(see Appendix A.3 for proof).*

Then, we show that with good probability, a point close to the optimal center of a good type-1 cluster is sampled. To prove this, we first establish the following the technical lemma.

**Lemma 6.** *Let* $L$ *be a set of lines in* $\mathbb{R}^d$, *and let* $c, c' \in \mathbb{R}^d$ *be two points. Then, the total squared distance from the lines in* $L$ *to* $c$ *satisfies:* $\Delta(L, \{c\}) \leq 3|L| \cdot d(c, c') + 6 \cdot \Delta(L, \{c'\})$, *where* $\Delta(L, \{c\}) = \sum_{\ell \in L} \delta(\ell, c)$ *and* $\delta(\ell, c)$ *denote the squared Euclidean distance from line* $\ell$ *to point* $c$, *respectively (see Appendix A.4 for the proof).*

This lemma, used in the proof of Lemma 6, shows that the clustering cost of any two centers is related to the distance between the centers. In the following lemma, we show that if a cluster with respect to $C$ incurs a large clustering cost, then the subset of lines located near its center likewise contributes a comparably high cost.

**Lemma 7.** *Let* $Q$ *be a set of lines in* $\mathbb{R}^d$, *let* $C \subseteq \mathbb{R}^d$ *be a set of* $k$ *centers, and let* $c^*$ *be the optimal single center for* $Q$. *Assume* $\alpha \geq 9$. *If the cost under* $C$ *satisfies* $\Delta(Q, C) \geq \alpha \cdot \Delta(Q, \{c^*\})$, *then there exists a subset* $R \subseteq Q$ *with* $\Delta(R, C) \geq \frac{\alpha-1}{24} \cdot \Delta(Q, \{c^*\})$, *where* $R = \{\ell \in Q \mid \delta(\ell, c^*) \leq \frac{2}{|Q|} \cdot \Delta(Q, \{c^*\})\}$ *(see appendix A.5 for proof).*

To find a point near the optimal center along a line, we first evaluate the probability of sampling a line close to the optimal center. For any $c \in C_S$, we have $\Delta(L(\psi_S(c)), C) \geq 9 \cdot \Delta(L(\psi_S(c)), \{\psi_S(c)\})$ due to Definition 3. Then, by applying Lemma 7 with $Q = L(\psi_S(c))$ and $\alpha = 9$, we get that $\Delta(\mathcal{A}, C) \geq \frac{\alpha-1}{24} \cdot \Delta(L(\psi_S(c)), \{\psi_S(c)\}) = \frac{\alpha-1}{24\alpha} \cdot \Delta(L(\psi_S(c)), C) \geq \frac{1}{27} \cdot \Delta(L(\psi_S(c)), C)$, where $\mathcal{A}$ is the set of lines within distance $\frac{2\Delta(L(\psi_S(c)), \{\psi_S(c)\})}{|L(\psi_S(c))|}$ from $\psi_S(c)$. By Lemma 5, we can obtain that $\sum_{c \in C_S \cap C_S^g} \Delta(L(\psi_S(c)), C) \geq \frac{1}{50} \cdot \Delta(L, C)$. Thus, we conclude that the total cost over all sets $\mathcal{A}$ from good clusters satisfies $\sum_{c \in C_S \cap C_S^g} \Delta(\mathcal{A}, C) \geq \frac{1}{27 \cdot 50} \cdot \Delta(L, C)$, implying that the probability of sampling a line from $\bigcup_{c \in C_S \cap C_S^g} \mathcal{A}$ is at least $\frac{1}{1500}$. However, since $\mathcal{A}$ is a set of lines rather than points, we cannot directly select a line as a candidate center. Instead, we need to generate a set of points from the lines in $\mathcal{A}$, and ensure that at least one of them lies within distance $\frac{2\Delta(L(\psi_S(c)), \{\psi_S(c)\})}{|L(\psi_S(c))|}$ from the optimal center. To proceed, we introduce a geometric discretization structure, referred to as $r$-*CrossLine*, and prove that the set of points returned by this structure contains at least one point that lies within distance $r$ of any point inside the structure.

**Definition 4** ($r$-*CrossLine* in $\mathbb{R}^2$). *Let* $\ell_1$ *and* $\ell_2$ *be two intersecting lines in* $\mathbb{R}^2$, *intersecting at a point* $p = \ell_1 \cap \ell_2$. *For each line* $\ell_i$ ($i = 1, 2$), *we construct two lines parallel to* $\ell_i$ *by shifting it along its normal direction by* $\pm r$. *Let* $\mathcal{L}_1 = \{\ell_1^-, \ell_1, \ell_1^+\}$ *and* $\mathcal{L}_2 = \{\ell_2^-, \ell_2, \ell_2^+\}$ *denote the sets of three parallel lines derived from* $\ell_1$ *and* $\ell_2$, *respectively. The* $r$-*CrossLine is defined as the union of all regions enclosed by the lines in* $\mathcal{L}_1$ *and* $\mathcal{L}_2$, *and* $CrossLine_r(\ell_1, \ell_2) = \{\ell_i \cap \ell_j \mid \ell_i \in \mathcal{L}_1, \ell_j \in \mathcal{L}_2\}$ *denotes the set of all pairwise intersection points.*

Based on Definition 4, we prove that for any point inside the $r$-*CrossLine*, there exists at least one grid point within distance $r$ of it.

**Lemma 8.** *Let* $\ell_1$ *and* $\ell_2$ *be two intersecting lines in* $\mathbb{R}^2$, *and let* $r > 0$ *be a constant. Then, for any point* $x$ *within* $r$-*CrossLine, there exists at least one grid point from* $CrossLine_r(\ell_1, \ell_2)$ *whose distance to* $x$ *is at most* $r$. *(see Appendix A.6 for proof).*

As mentioned earlier, $\mathcal{A}$ is the set of lines within distance $\frac{2\Delta(L(\psi_S(c)), \{\psi_S(c)\})}{|L(\psi_S(c))|}$ from $\psi_S(c)$, and the probability of sampling a line from the set $\bigcup_{c \in C_S \cap C_S^g} \mathcal{A}$ is at least $\frac{1}{1500}$. Therefore, the probability that both $\ell_1$ and $\ell_2$ lie in $\mathcal{A}$ is at least $\zeta = \left(\frac{1}{1500}\right)^2$. Assuming that $\ell_1, \ell_2 \in \bigcup_{c \in C_S \cap C_S^g} \mathcal{A}$, $r = \frac{2\Delta(L(\psi_S(c)), \{\psi_S(c)\})}{|L(\psi_S(c))|}$, let $CrossLine_r(\ell_1, \ell_2)$ be the point set obtained by Definition 4 based on $\ell_1$

and $\ell_2$. Then, by Lemma 8, *CrossLine*$_r(\ell_1, \ell_2)$ contains at least one point within distance $r$ of the optimal center, which can be used as a high-quality swap candidate. However, since the radius $r$ is defined in terms of the (unknown) optimal cost, we adopt a multi-scale, data-driven schedule: let $r_{\min}$ and $r_{\max}$ be empirical scales around the intersection (e.g., lower/upper quantiles of the distances from the intersection to the candidate set returned by Algorithm 2). We progressively explore the search radii $r_t \in \left[\frac{r_{\min}}{10}, r_{\max}\right]$, constructing an $r_t$-*CrossLine* around the intersection at each scale and performing a single-swap at every step until an improvement is found. By construction, at least one of the radii $r_t$ lies within a constant factor of the theoretical radius $r$. At this scale, Lemma 8 guarantees a constant probability of success. By Definition 3, if such a point $p$ is obtained, we can swap it with $c$ to get a new clustering with cost at most $\Delta(L, C \setminus \{c_h\} \cup \{p\}) \leq \Delta(L, C) - \Delta(L(\psi_S(c)), C) + \eta_1(c) + \Delta(L(\psi_S(c)), \{c\})$. By Lemma 6, we have that $\Delta(L(\psi_S(c)), \{c\}) \leq 9 \cdot \Delta(L(\psi_S(c)), \{\psi_S(c)\})$. Thus, with a constant probability, the new clustering has cost at most $\Delta(L, C) - (\Delta(L(\psi_S(c)), C) - \eta_1(c) - 9\Delta(L(\psi_S(c)), \{\psi_S(c)\})) \leq \left(1 - \frac{1}{100k}\right) \cdot \Delta(L, C)$.

### 3.2.2 The Analysis of Case 2

For type-2 matched swap pair $(c, \psi_N(c))$ $(c \in C_N)$, we similarly define clusters that yield a significant reduction in assignment cost under such swaps as *good clusters*, as formalized below.

**Definition 5** (*Good type-2 cluster*). *For any $c \in C_N$, the cluster $L(c)$ is called a good type-2 cluster, if $\Delta(L(\psi_N(c)), C) - \eta_2(c) - 9\Delta(L(\psi_N(c)), \{\psi_N(c)\}) > \frac{1}{100k}\Delta(L, C)$. Otherwise, $L(c)$ is called a bad type-2 cluster with $(c, \psi_N(c))$.*

For simplicity, let $C_N^g$ denote the set of centers in $C_N$ for which the corresponding cluster is a good type-2 cluster. Let $C_N^b = C_N \backslash C_N^g$. The above definition estimates the cost reduction of replacing $c$ with a point close to $\psi_S(c)$ by considering a cluster that reassigns the lines in $L(c)$ and assigns all lines in $L(\psi_S(c))$ to the new center. We now aim to show that a good type-2 cluster can be sampled with high probability. To this end, we first argue that the total cost contributed by good type-2 clusters is sufficiently large.

**Lemma 9.** *If $\sum_{c \in C \backslash C_S} \Delta(L(\psi_N(c)), C) \geq \frac{2}{3} \cdot \Delta(L, C)$, then we have $\sum_{c \in C \backslash C_S, c \in C_N^g} \Delta(L(\psi_N(c)), C) \geq \frac{1}{50} \cdot \Delta(L, C)$ (see Appendix A.7 for proof).*

Note that in this case we can now argue similarly as in the other case that sampling according to sum of squared distances will provide us with constant probability with a good center using Lemma 7. Indeed, since the sum of squared distances of lines in good centers is at least $\frac{1}{50}\Delta(L, C)$ by Lemma 9, it follows together with Lemma 7 that we sample a point from a good cluster $L(\phi_N(c))$ that is within distance two times the average cost of the cluster with probability $\frac{1}{1500}$. By the definition of good cluster, we have that such a point improves the cost of the current clustering by at least a factor of $\left(1 - \frac{1}{100k}\right)$. Thus, by combining Case 1 and Case 2, we complete the proof of Lemma 3. Now we complete the proof of Theorem 1. Let $\hat{C}$ be the set of centers returned by step 2 of SLS-$k$-ML, which achieves a $\rho$-approximation guarantee by Lemma 2. Let $C$ be the set of centers returned by SLS-K-ML. By Lemma 3, we know that if the current cost exceeds $500\tau$, then with probability at least $\Omega(1/\rho)$, a single iteration of SLS-K-ML reduces the cost by a factor of $\left(1 - \frac{1}{100k}\right)$. We introduce a random process $X$ to analyze the change of the clustering cost. The process starts at $\Delta(L, \hat{C})$, and over $T = 100k\zeta \log \rho$ steps, it decreases multiplicatively by a factor of $\left(1 - \frac{1}{100k}\right)$ with probability $1/\zeta$ at each step, and finally increases additively by $500\tau$. Furthermore, $\mathbb{E}[X] = 500\tau + \Delta(L, \hat{C}) \cdot \sum_{i=0}^{T} \binom{T}{i}(\frac{1}{\zeta})^i(\frac{\zeta-1}{\zeta})^{T-i}(1 - \frac{1}{100k})^i = \Delta(L, \hat{C})(1 - \frac{1}{100k\zeta})^{100k\zeta \log \rho} + 500\tau \leq \frac{\Delta(L, \hat{C})}{\rho} + 500\tau$. This implies that $\mathbb{E}[\Delta(L, C) \mid \hat{C}] \leq \frac{\Delta(L, \hat{C})}{\rho} + 500\tau$. Then, we have $\mathbb{E}[\Delta(L, C)] = \sum_{\hat{C}} \mathbb{E}[\Delta(L, C) \mid \hat{C}] \cdot \Pr(\hat{C}) = \sum_{\hat{C}} \Pr(\hat{C})(\frac{\Delta(L, \hat{C})}{\rho} + 500\tau) \leq (500 + \varepsilon)\tau$.

**Running Time Analysis.** The overall time complexity of SLS-$k$-ML is $O(n^2 + nk^2\zeta \log \rho)$. The first stage of constructing the initial solution takes $O(n^2)$ time, as it involves constructing the candidate set using the CENTROID-SET procedure, which takes $O(n^2)$ time, and sampling $k$ points from the candidate set, which takes $O(1)$ time. To achieve a $(500 + \varepsilon)$-approximate solution, SLS-$k$-ML requires $O(k\zeta \log \rho)$ iterations. In each iteration, sampling two lines takes $O(\log n)$ time, constructing the *CrossLine* structure takes $O(1)$ time, finding swap points from the *CrossLine* output takes $O(1)$ time, and updating the distances between all lines and their nearest centers takes $O(nk)$

time. Therefore, the total running time of the iterative procedure is $O(nk^2\zeta \log \rho)$, and the overall time complexity of SLS-$k$-ML is $O(n^2 + nk^2\zeta \log \rho)$.

## 3.3 Extended to $\mathbb{R}^d$

In this section, we show how to extend our algorithm to the general case in $\mathbb{R}^d$. In $\mathbb{R}^d$, the the existence of skew lines makes the *CrossLine* structure in $\mathbb{R}^2$ inadequate, as lines may not intersect and can lie on different affine subspaces. To address this, we redefine the *CrossLine* structure to accommodate high-dimensional geometry, ensuring coverage guarantees remain valid under arbitrary line orientations. Moreover, in $\mathbb{R}^d$, the reassignment argument remains valid, while the bound must be adjusted to account for the dimensional dependence in the projection geometry. Formally, we restate the following result for $\mathbb{R}^d$.

**Lemma 10.** *For any $c \in C_S$ (or $c \in C_N$), the reassignment cost $\eta_1(c)$ (or $\eta_2(c)$) is at most $\frac{1}{5}\Delta(L(c), C) + 24\Delta(L(c), C^*)$ (see Appendix A.8 for proof).*

Note that although Lemma 4 and Lemma 10 yield the same conclusion, they are derived in different contexts and serve distinct purposes within the analysis. The second difference lies in the construction of the *CrossLine* structure. Recall that in $\mathbb{R}^2$, we generate a constant-size candidate set based on the intersection and perturbation of line pairs. In $\mathbb{R}^d$, this structure must be extended to cover a higher-dimensional geometric region induced by two non-parallel lines. Specifically, for each pair of lines, we construct a discretized grid over the intersection of their respective $r$-bounded neighborhoods, which forms a $d$-dimensional crossline region. Note that in the process of constructing the candidate set in $\mathbb{R}^d$, we retain the same discretization resolution as in the planar case, but extend it across all $d$ orthogonal directions to ensure sufficient coverage for sampling high-quality swap points. We now introduce the required extensions to the candidate set construction.

**Definition 6** (*$r$-CrossLine* in $\mathbb{R}^d$). *Let $\ell_1, \ell_2 \subset \mathbb{R}^d$ be two non-parallel lines. For each line $\ell_i$ ($i = 1, 2$), we define the set of $3^d$ axis-aligned parallel shifts as $\mathcal{L}_i = \{\ell_i^{\vec{s}} = \ell_i + \sum_{j=1}^d s_j \cdot r \cdot e_j \mid \vec{s} \in \{-1, 0, 1\}^d\}$, where $e_j$ is the unit vector in the $j$-th coordinate direction. We define the $r$-CrossLine as the union of all regions enclosed by the lines in $\mathcal{L}_1$ and $\mathcal{L}_2$, and $CrossLine_r(\ell_1, \ell_2) = \{\ell \cap \ell' \mid \ell \in \mathcal{L}_1, \ell' \in \mathcal{L}_2, \ell \cap \ell' \neq \emptyset\}$. Assuming each pair of lines $(\ell, \ell')$ intersects (e.g., due to careful shift alignment in a common plane), the total number of grid points is at most $|CrossLine_r(\ell_1, \ell_2)| \leq 3^{2d}$.*

Based on the $r$-CrossLine in $\mathbb{R}^d$, we restate Lemma 8 as follows.

**Lemma 11.** *Let $\ell_1$ and $\ell_2$ be two intersecting lines in $\mathbb{R}^d$, and let $r > 0$ be a constant. Then, for any point $x$ within $r$-CrossLine, there exists at least one grid point from $CrossLine_r(\ell_1, \ell_2)$ whose distance to $x$ is at most $r$ (see Appendix A.9 for the proof).*

**Running Time Analysis.** The overall time complexity of SLS-$k$-ML is $O(n^2d^2 + 9^d ndk^2\zeta \log \rho)$. The first stage of constructing the initial solution takes $O(n^2d^2)$ time, as it involves constructing the candidate set using the CENTROID-SET procedure, which takes $O(n^2d^2)$ time, and sampling $k$ points from the candidate set, which takes $O(1)$ time. To achieve a $(500 + \varepsilon)$-approximate solution, SLS-$k$-ML requires $O(k\zeta \log \rho)$ iterations. In each iteration, sampling two lines takes $O(\log n)$ time, constructing the *CrossLine* structure takes $O(1)$ time, finding swap points from the *CrossLine* output takes $O(9^d)$ time, and updating the distances between all lines and their nearest centers takes $O(ndk)$ time. Therefore, the total running time of the iterative procedure is $O(9^d ndk^2\zeta \log \rho)$, and the overall time complexity of SLS-$k$-ML is $O(n^2d^2 + 9^d nk^2\zeta \log \rho)$.

## 4 Experiments

In this section, we give empirical evaluations on the performances of our proposed algorithms. All algorithms are implemented and executed in Python. The experiments were done on a machine with i7-14700KF processor and 256GB RAM.

**Datasets.** We evaluate the performance of our algorithm on both synthetic datasets (SYN 1 with $n = 5000$, $d = 10$, and SYN2 with $n = 10000$, $d = 5$) and real-world OpenStreetMap datasets (RE 1 with $n = 476$, $d = 2$, and RE 2 with $n = 418$, $d = 2$) as used in [20]. For each dataset, we run both algorithms 10 times and report the minimum cost (*Min_Cost*), maximum cost (*Max_Cost*), average cost (*Avg_Cost*), standard deviation (*Std*), and runtime (*Time(s)*).

Table 1: Experimental results of our SLS-$k$-ML algorithm and the coreset-based method.

| Datasets | Method | $k$ | Min_Cost | Max_Cost | Avg_Cost | Std | Time(s) |
|---|---|---|---|---|---|---|---|
| RE 1 | SLS-$k$-ML(Ours) | 10 | **8.37E-10** | **1.72E-06** | **1.01E-07** | **2.73E-07** | **1.05** |
| | Coreset+sampling | 10 | 8.17E-04 | 3.08E-02 | 6.24E-03 | 5.89E-03 | 6.05 |
| | Coreset+exhaustive search | 10 | - | - | - | - | Over 12 hours |
| RE 1 | SLS-$k$-ML(Ours) | 3 | **5.67E-06** | **1.15E-03** | **1.35E-04** | **2.00E-04** | **0.71** |
| | Coreset+sampling | 3 | 1.57E-02 | 8.78E-01 | 1.39E-01 | 1.95E-01 | 4.08 |
| | Coreset+exhaustive search | 3 | 1.10E-03 | 2.11E-03 | 1.56E-03 | 3.01E-04 | 93.97 |
| RE 2 | SLS-$k$-ML(Ours) | 10 | **1.35E-08** | **3.16E-05** | **1.60E-06** | **4.66E-06** | **1.40** |
| | Coreset+sampling | 10 | 2.89E-03 | 1.20E-01 | 2.16E-02 | 2.13E-02 | 3.34 |
| | Coreset+exhaustive search | 10 | - | - | - | - | Over 12 hours |
| RE 2 | SLS-$k$-ML(Ours) | 3 | **1.45E-05** | **5.59E-03** | **1.39E-03** | **1.18E-03** | **0.43** |
| | Coreset+sampling | 3 | 4.88E-02 | 6.11E+00 | 6.21E-01 | 1.08E+00 | 4.91 |
| | Coreset+exhaustive search | 3 | 4.69E-04 | 4.83E-03 | 2.51E-03 | 1.09E-03 | 2.59E+02 |
| SYN 1 | SLS-$k$-ML(Ours) | 10 | **1.84E+04** | **1.88E+04** | **1.86E+04** | **1.97E+02** | **4.56E+02** |
| | Coreset+sampling | 10 | 3.97E+04 | 4.51E+04 | 4.24E+04 | 2.70E+03 | 2.04E+04 |
| | Coreset+exhaustive search | 10 | | | | | Over 12 hours |
| SYN 2 | SLS-$k$-ML(Ours) | 3 | **1.87E+04** | **1.98E+04** | **1.98E+04** | **5.45E+02** | **1.15E+03** |
| | Coreset+sampling | 3 | - | - | - | - | Over 12 hours |
| | Coreset+exhaustive search | 3 | - | - | - | - | Over 12 hours |

**Algorithms.** In our experiments, we give comparisons between our local search algorithm and the coreset-based method from [20]. For coreset-based method, we compress the data with their coreset algorithm and select centers via sampling or exhaustive search. For our SLS-$k$-ML algorithm, we design a sampling strategy that selects a subset of 100 points from $r$-*CrossLine* to improve computational efficiency. Following the settings in [14], we set the number of sampling rounds to $T = 400$, and the number of clusters to $k = 3$ and $k = 10$.

**Results.** Table 1 shows that our proposed SLS-$k$-ML algorithm always achieve the best performance compared with coreset+sampling algorithm and coreset+exhaustive algorithm. In particular, on datasets containing more than 5000 input lines, our algorithm runs at least 43 times faster while maintaining or improving clustering quality. Furthermore, the consistently low standard deviation observed across all experiments demonstrates the robustness of our algorithm.

## 5  Conclusions

In this paper, we propose the first local search algorithm for the $k$-means of lines problem, based on a single-swap strategy, which achieves a $(500 + \varepsilon)$-approximation guarantee and runs in polynomial time for low-dimensional Euclidean space. To handle the lack of triangle inequality and the geometric complexity of line clustering, we design two core components: a *proportional capture relation* for aligning optimal and current centers, and a *CrossLine* structure for discretizing line interactions. Extensive experiments on both synthetic and real-world datasets demonstrate that our algorithm consistently outperforms coreset-based baselines in both efficiency and clustering quality. However, our current theoretical analysis and experimental evaluation are mainly limited to low-dimensional settings, and alleviating the $d$-dependence will be an interesting direction for future work.

## Acknowledgments and Disclosure of Funding

This work was supported by the National Natural Science Foundation of China (Nos. 62432016, 62172446) and the Central South University Research Program of Advanced Interdisciplinary Studies (No. 2023QYJC023). This work was also carried out in part using computing resources at the High Performance Computing Center of Central South University.

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

# A The Missing Proof

## A.1 Proof of Lemma 2

**Lemma 2.** *Given an instance $\mathcal{I} = (L, d, k)$ of the $k$-ML problem, let $C$ be the set of points returned by step 2 of SLS-$k$-ML. Then, for a finite constant $\rho$, we have $\Delta(L, C) \leq \rho \cdot \Delta(L, C^*)$.*

*Proof.* Let $C^* = \{c_1^*, \ldots, c_k^*\}$ denote the optimal solution of $\mathcal{I}$. For any $\ell \in L$, let $c \in C$ and $c^* \in C^*$ denote the closest centers to $\ell$ in $C$ and $C^*$, respectively. Then, we have

$$
\begin{aligned}
\Delta(L, C) = \sum_{\ell \in L} \delta(\ell, c) &\leq \sum_{\ell \in L} \left( \sqrt{\delta(\ell, c^*)} + \sqrt{\delta(\pi_\ell(c), \delta(\pi_\ell(c^*))} + \sqrt{\delta(c, c^*)} \right)^2 \\
&\leq \sum_{\ell \in L} \left( \sqrt{\delta(\ell, c^*)} + 2\sqrt{\delta(c, c^*)} \right)^2 \\
&\leq \sum_{\ell \in L} \delta(\ell, c^*) + 4\delta(c, c^*) + 4\sqrt{\delta(\ell, c^*)}\sqrt{\delta(c, c^*)} \\
&\leq \sum_{\ell \in L} 3\delta(\ell, c^*) + 6\delta(c, c^*),
\end{aligned}
$$

where the first inequality follows from triangle inequality, the second step follow from the fact that the distance between the projections of two points onto a line is at most their original Euclidean distance, and the last step follows the squared inequality and Cauchy inequality, respectively. Based on the Theorem in [24], the optimal solution to the $k$-means of lines problem can be obtained by selecting centers from the set of all pairwise line intersections (in $\mathbb{R}^2$). Since the maximum pairwise distance between any two points in the candidate set $P$ can be explicitly computed, we have $\Delta(L, C) \leq \rho \cdot \Delta(L, C^*)$, where $\rho$ is a constant. Moreover, the initial solution can be obtained in $O(n^2 d^2)$ time, as constructing the candidate set $P$ takes $O(n^2 d^2)$ time and sampling $k$ centers from $P$ takes $O(1)$ time. $\qquad\square$

## A.2 Proof of Lemma 4

**Lemma 4.** *For any $c \in C_S$ (or $c \in C_N$), the reassignment cost $\eta_1(c)$ (or $\eta_2(c)$) is at most $\frac{1}{5}\Delta(L(c), C) + 24\Delta(L(c), C^*)$.*

*Proof.* We first consider a type-1 match swap pair $(c, \psi_S(c))$ ($c \in C_S$), as the case of type-2 match swap pair is nearly identical and even simpler. Let $c^* = \psi_S(c)$. By definition 2, we have $\eta_1(c) = \Delta(L \backslash L(c^*), C \backslash \{c\}) - \Delta(L \backslash L(c^*), C)$, since lines in clusters other than $L(c)$ will still be assigned to their current center. Consider a line $\ell \in L(c) \backslash L(c^*)$, and it is easy to get that $\ell$ does not belong to the optimal cluster $L(c^*)$. Let $c_\ell^*$ ($c_\ell^* \neq c^*$) denote the closet optimal center in $C^*$ from $\ell$. Then, we construct a line $\ell'$ that is the translation of $\ell$ to $c_\ell^*$, and then find a point $c_p$ that is the closet center in $C$ to $\ell'$. It is easy to get that $c_p \neq c$, otherwise $\ell$ will remain assigned to its original center and no reassignment will be necessary. We now assign every lines in $L(c) \backslash L(c^*)$ to $c_p$, and get an estimate for the cost of reassigning these lines. Figure 1(b)-(c) shows two configurations of $\ell, c, c_\ell^*$, and $c_p$. From Figure 1(b), we have

$$
\begin{aligned}
\eta_1(c) \leq \sum_{\ell \in L(c) \backslash L(c^*)} (\delta(\ell, c_p) - \delta(\ell, c)) &\leq \sum_{\ell \in L(c) \backslash L(c^*)} \left( \sqrt{\delta(\ell, c_\ell^*)} + \sqrt{\delta(\ell', c_p)} \right)^2 - \delta(\ell, c) \\
&\leq \sum_{\ell \in L(c) \backslash L(c^*)} \left( 2\sqrt{\delta(\ell, c_\ell^*)} + \sqrt{\delta(\ell, c)} \right)^2 - \delta(\ell, c) \\
&\leq \sum_{\ell \in L(c) \backslash L(c^*)} 4\delta(\ell, c_\ell^*) + 2\sqrt{\frac{2}{\lambda}} \cdot \sqrt{2\lambda \cdot \delta(\ell, c_\ell^*) \cdot \delta(\ell, c)} \\
&\leq \sum_{\ell \in L(c) \backslash L(c^*)} (4 + \frac{2}{\lambda}) \cdot \delta(\ell, c_\ell^*) + 2\lambda \cdot \delta(\ell, c),
\end{aligned}
$$

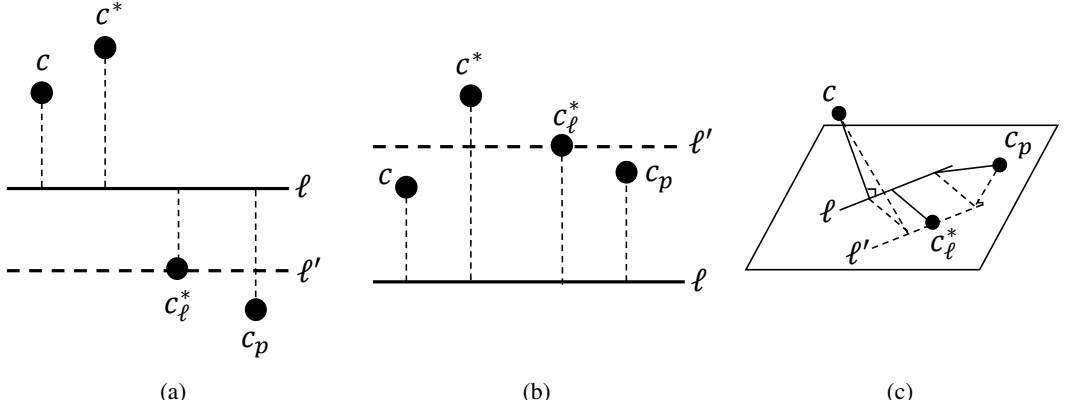

Figure 1: Illustrations of Line Configurations in $\mathbb{R}^2$ and $\mathbb{R}^d$.

where the first inequality follows from equidistance property of parallel lines and squared inequality, the second step follow from the fact that $c_p$ is the closet center in $C$ to $\ell'$, and the last step follows from the Cauchy Inequality, respectively. From Figure 1(c), we have

$$
\eta_1(c) \leq \sum_{\ell \in L(c) \backslash L(c^*)} (\delta(\ell, c_p) - \delta(\ell, c)) \leq \sum_{\ell \in L(c) \backslash L(c^*)} \left( 2\sqrt{\delta(\ell, c_\ell^*)} + \sqrt{\delta(\ell, c)} \right)^2 - \delta(\ell, c)
$$

$$
\leq \sum_{\ell \in L(c) \backslash L(c^*)} 4\delta(\ell, c_\ell^*) + 2\sqrt{\frac{2}{\lambda}} \cdot \sqrt{2\lambda \cdot \delta(\ell, c_\ell^*) \cdot \delta(\ell, c)}
$$

$$
\leq \sum_{\ell \in L(c) \backslash L(c^*)} (4 + \frac{2}{\lambda}) \cdot \delta(\ell, c_\ell^*) + 2\lambda \cdot \delta(\ell, c),
$$

where the first inequality follows the fact that $\delta(\ell, c_p) < \delta(\ell, c_\ell^*)$ under the configuration of Figure 1(c), the second steps and the last step follows from squared inequality and Cauchy Inequality. For the configurations of Figure 1(b)-(c), let $\lambda = \frac{1}{10}$. Then $\eta_1(c) \leq \frac{1}{5}\Delta(L(c), C) + 24\Delta(L(c), C^*)$. The cases for type-2 match swap pair $(c, \psi_N(c))$ $(c \in C_N)$ are similar to the cases for type-1 match swap pair. $\qquad\square$

### A.3 Proof of Lemma 5

**Lemma 5.** *If* $3 \sum_{c \in C_S} \Delta(L(\psi_S(c)), C) > \Delta(L, C)$*, then we have* $\sum_{c \in C_S \cap C_S^g} \Delta(L(\psi_S(c)), C) \geq \frac{1}{50} \cdot \Delta(L, C)$.

*Proof.* By Definition 3 and Lemma 4, we have

$$
\sum_{c \in C_S \cap C_S^b} \Delta(L(\phi_S(c)), C) \leq \sum_{c \in C_S} \eta_1(c) + 9 \cdot \tau + \frac{1}{100} \cdot \Delta(L, C) \leq \frac{21}{100} \cdot \Delta(L, C) + 33 \cdot \tau.
$$

Based on the assumption that $\Delta(L, C) \geq 500 \cdot \tau$, we obtain $\sum_{c \in C_S \cap C_S^b} \Delta(L(\phi_S(c)), C) \leq \frac{69}{250} \cdot \Delta(L, C)$, and thus $\sum_{c \in C_S \cap C_S^b} \Delta(L(\phi_S(c)), C) \geq \frac{3}{50} \cdot \Delta(L, C)$. $\qquad\square$

### A.4 Proof of Lemma 6

**Lemma 6.** *Let $L$ be a set of lines in $\mathbb{R}^d$, and let $c, c' \in \mathbb{R}^d$ be two points. Then, the total squared distance from the lines in $L$ to $c$ satisfies:* $\Delta(L, \{c\}) \leq 3|L| \cdot d(c, c') + 6 \cdot \Delta(L, \{c'\})$*, where* $\Delta(L, \{c\}) = \sum_{\ell \in L} \delta(\ell, c)$ *and* $\delta(\ell, c)$ *denote the squared Euclidean distance from line $\ell$ to point $c$, respectively.*

*Proof.* For any point $c \in \mathbb{R}^2$ and line $\ell \in Q$, let $\pi_\ell(c)$ denote the projection of $c$ on $\ell$. let $c$ be a center in $C$. We have

$$\Delta(L, \{c\}) \leq \sum_{\ell \in L} \delta(\ell, c) \leq \sum_{\ell \in L} \left( \sqrt{\delta(\ell, c^*)} + \sqrt{\delta(\pi_\ell(c), \pi_\ell(c^*))} + \sqrt{\delta(c, c^*)} \right)^2$$

$$\leq \sum_{\ell \in L} \left( \sqrt{\delta(\ell, c^*)} + 2\sqrt{\delta(c, c^*)} \right)^2$$

$$\leq \sum_{\ell \in L} \delta(\ell, c^*) + 4\delta(c, c^*) + 4\sqrt{\delta(\ell, c^*)}\sqrt{\delta(c, c^*)}$$

$$\leq \sum_{\ell \in L} 3\delta(\ell, c^*) + 6\delta(c, c^*)$$

$$\leq 3\Delta(L, \{c^*\}) + 6|L|\delta(c, c^*)$$

where the first inequality follows from the triangle inequality, the second step follow from the fact that the distance between the projections of two points onto a line is at most their original Euclidean distance, respectively. $\square$

## A.5 Proof of Lemma 7

**Lemma 7.** *Let $Q$ be a set of lines in $\mathbb{R}^d$, let $C \subseteq \mathbb{R}^d$ be a set of $k$ centers, and let $c^*$ be the optimal single center for $Q$. Assume $\alpha \geq 9$. If the cost under $C$ satisfies $\Delta(Q, C) \geq \alpha \cdot \Delta(Q, \{c^*\})$, then there exists a subset $R \subseteq Q$ with $\Delta(R, C) \geq \frac{\alpha-1}{24} \cdot \Delta(Q, \{c^*\})$, where $R = \{\ell \in Q \mid \delta(\ell, c^*) \leq \frac{2}{|Q|} \cdot \Delta(Q, \{c^*\})\}$.*

*Proof.* Based on the Lemma 6, we know that the closest center in $C$ to $c^*$ has distance at least $\frac{\alpha-3}{6|Q|} \cdot \Delta(Q, \{c^*\})$ as otherwise $\Delta(Q, C) < \alpha \cdot \Delta(Q, \{c^*\})$. Hence, the distance of every lines in $R$ to $C$ is at least

$$\left( \sqrt{2} - 2\sqrt{\frac{\alpha-3}{6}} \right)^2 \cdot \frac{\text{cost}(Q, \{c^*\})}{|Q|} \geq \frac{\alpha-1}{24} \cdot \frac{\text{cost}(Q, \{c^*\})}{|Q|},$$

where we use that $\alpha \geq 9$. Furthermore, by averaging we get $|R| \geq |Q|/2$, which together with the inequality above implies the result. $\square$

## A.6 Proof of Lemma 8

**Lemma 8.** *Let $\ell_1$ and $\ell_2$ be two intersecting lines in $\mathbb{R}^2$, and let $r > 0$ be a constant. Then, for any point $x$ within $r$-CrossLine, there exists at least one grid point from $CrossLine_r(\ell_1, \ell_2)$ whose distance to $x$ is at most $r$.*

*Proof.* The 9 grid points defined in $\text{CrossLine}_r(\ell_1, \ell_2)$ form a regular $3 \times 3$ grid in a parallelogram arrangement. The lines in $\mathcal{L}_1$ are equally spaced at distance $r$, and so are the lines in $\mathcal{L}_2$. Thus, the grid divides the local region around the intersection point into parallelogram cells, each defined by two adjacent lines from $\mathcal{L}_1$ and two from $\mathcal{L}_2$. Any point $x$ within $\text{CrossLine}_r(\ell_1, \ell_2)$ must lie in one of these parallelogram cells. Since each parallelogram has diameter (i.e., maximal corner-to-center distance) less than or equal to $r$, any point $x$ must be within distance $r$ of at least one vertex of the cell, which is a grid point of $\text{CrossLine}_r(\ell_1, \ell_2)$. Hence, the lemma follows. $\square$

## A.7 Proof of Lemma 9

**Lemma 9.** *If $\sum_{c \in C \setminus C_S} \Delta(L(\psi_N(c)), C) \geq \frac{2}{3} \cdot \Delta(L, C)$, then we have $\sum_{c \in C \setminus C_S, c \in C_N^g} \Delta(L(\psi_N(c)), C) \geq \frac{1}{50} \cdot \Delta(L, C)$.*

*Proof.* Note that $|C\backslash C_S| \leq 2|C_N|$. By the Definition 5 and Lemma 4, we have

$$\sum_{c\in C\backslash C_S, c\in C_N^b} \Delta(L(\psi_N(c)), C) \leq 2|C_N| \min_{c\in C_N} \eta_2(c) + 9\tau + \frac{1}{100}\Delta(L, C)$$

$$\leq 2\sum_{c\in C_N} \eta_2(c) + 9\tau + \frac{1}{100}\Delta(L, C)$$

$$\leq \frac{41}{100}\Delta(L, C) + 57\tau.$$

Based on the assumption that $\Delta(L, C) \geq 500 \cdot \tau$, we obtain $\sum_{c\in C\backslash C_S, c\in C_N^b} \Delta(L(\psi_N(c)), C) \leq \frac{131}{250} \cdot \Delta(L, C)$, and thus $\sum_{c\in C\backslash C_S, c\in C_N^g} \Delta(L(\phi_N(c)), C) \geq \frac{7}{50} \cdot \Delta(L, C)$. $\square$

### A.8 Proof of Lemma 10

**Lemma 10.** *For any $c \in C_S$ (or $c \in C_N$), the reassignment cost $\eta_1(c)$ (or $\eta_2(c)$) is at most $\frac{1}{5}\Delta(L(c), C) + 24\Delta(L(c), C^*)$ (see Appendix A.8 for proof).*

*Proof.* Similar to $\mathbb{R}^d$, we also construct a line $\ell'$ that is the translation of $\ell$ to $c_\ell^*$, and then find a point $c_p$ that is the closet center in $C$ to $\ell'$. We now assign every lines in $L(c)\backslash L(c^*)$ to $c_p$, and get an estimate for the cost of reassigning these lines. From Figure 1(c), we have

$$\eta_1(c) \leq \sum_{\ell\in L(c)\backslash L(c^*)} (\delta(\ell, c_p) - \delta(\ell, c)) \leq \sum_{\ell\in L(c)\backslash L(c^*)} \left(\sqrt{\delta(\ell, c_\ell^*)} + \sqrt{\delta(\ell', c_p)}\right)^2 - \delta(\ell, c)$$

$$\leq \sum_{\ell\in L(c)\backslash L(c^*)} \left(2\sqrt{\delta(\ell, c_\ell^*)} + \sqrt{\delta(\ell, c)}\right)^2 - \delta(\ell, c)$$

$$\leq \sum_{\ell\in L(c)\backslash L(c^*)} 4\delta(\ell, c_\ell^*) + 2\sqrt{\frac{2}{\lambda}} \cdot \sqrt{2\lambda \cdot \delta(\ell, c_\ell^*) \cdot \delta(\ell, c)}$$

$$\leq \sum_{\ell\in L(c)\backslash L(c^*)} \left(4 + \frac{2}{\lambda}\right) \cdot \delta(\ell, c_\ell^*) + 2\lambda \cdot \delta(\ell, c),$$

where the first inequality follows from equidistance property of parallel lines and triangle inequality, the second step follow from the fact that $c_p$ is the closet center in $C$ to $\ell'$, and the last step follows from the Cauchy Inequality, respectively. For the configuration of Figure 1(c), let $\lambda = \frac{1}{10}$. Then $\eta_1(c) \leq \frac{1}{5}\Delta(L(c), C) + 24\Delta(L(c), C^*)$. The cases for type-2 match swap pair $(c, \psi_N(c))$ ($c \in C_N$) are similar to the cases for type-1 match swap pair. The lemma follows. $\square$

### A.9 Proof of Lemma 11

**Lemma 11.** *Let $\ell_1$ and $\ell_2$ be two intersecting lines in $\mathbb{R}^d$, and let $r > 0$ be a constant. Then, for any point $x$ within $r$-CrossLine, there exists at least one grid point from $CrossLine_r(\ell_1, \ell_2)$ whose distance to $x$ is at most $r$.*

*Proof.* By construction, each $\mathcal{L}_i$ contains $3^d$ lines, generated by shifting $\ell_i$ along all combinations of $\{-1, 0, 1\}^d$ at step size $r$ along coordinate axes. This induces a discrete grid of intersection points, where each point lies at the intersection of one line from $\mathcal{L}_1$ and one from $\mathcal{L}_2$. The resulting grid forms a regular structure embedded within the union of two axis-aligned hypercubes of side length $2r$, centered near a common point $x^*$. Since each axis-aligned hypercube is of side length $2r$, the distance between adjacent grid points along any axis is at most $r$, and each hypercube cell (in which grid points are located at vertices) has diameter at most $\sqrt{d} \cdot r$. Then, for any point $x$ lying within the convex hull of the grid points (i.e., the r-CrossLine region), there exists a hypercube cell that contains $x$. By geometry of axis-aligned cubes, any point within such a cell lies within distance $r$ from at least one of its $2^d$ corners, all of which are grid points. Hence, the lemma follows. $\square$

