# OpenReview forum: "A  Single-Swap Local Search Algorithm for k-Means of Lines"
_NeurIPS.cc/2025/Conference — NeurIPS 2025 poster_

### Official Review · Reviewer_N5Cm · 2025-06-28

**Clarity:** 4
**Significance:** 3
**Originality:** 4
**Rating:** 4
**Confidence:** 2

**Summary:**

The paper introduces an algorithm for clustering lines into k sets. The method is reminiscent of K-means++, but instead of sampling centers from a distribution, the authors discretize the continuous space using a grid and search over a finite set of candidate points. The theoretical analysis is solid and provides meaningful guarantees. I didn’t go through every proof in detail, but the parts I did check seemed correct, and the overall reasoning appears sound.

One limitation is the absence of numerical experiments—not even simple ones on synthetic data. While the authors mention that the method isn’t expected to be robust to noise, it would still be helpful to see the algorithm in action to better understand its behavior. The paper is clearly written, and although it’s more theoretical than most NeurIPS submissions, I think it’s a worthwhile contribution. That said, theory papers can be difficult to fully evaluate under conference reviewing timelines, and small technical issues might go unnoticed.

**Questions:**

Could you explain why you chose not to include any numerical experiments?
Even basic examples—successful or not—would help make the paper more accessible to a broader audience.

**Ethical Concerns:**

["NO or VERY MINOR ethics concerns only"]

**Final Justification:**

The paper presents a theoretical analysis of a k-means-for-lines algorithm. While the algorithm itself appears simple, it is both novel and interesting. The work is entirely theoretical, which was somewhat surprising; however, on request, the authors provided numerical results that met my original expectations. I believe the paper is stronger as a theory-only contribution, so I would not recommend including these results in the camera-ready version—though they could be a useful addition to the supplementary material.

Although I generally prefer to see applications alongside theory, I appreciate the authors’ clear and accessible exposition within the NeurIPS formatting constraints. Their approach emphasizes explanation over impressiveness for its own sake, which is commendable. Discussions with other reviewers and the authors’ rebuttal have reinforced my positive impression. I am therefore leaning towards acceptance.

**Limitations:**

Lack of empirical evaluation.
The analysis is based on idealized data, with no consideration for noise, although this is addressed in the discussion.
Potential sensitivity to initialization or local optima is not addressed.

**Paper Formatting Concerns:**

I did not notice anything

**Quality:**

3

**Strengths And Weaknesses:**

Strengths:
>Clear and well-organized theoretical analysis.
>Tackles a relatively unexplored problem.
>The algorithm design is interesting and non-trivial.

Weaknesses:
>No experimental validation.
>No discussion of robustness or practical performance.

---

> ### Author Rebuttal · Authors · 2025-07-31
>
> **Weakness and Question: No experimental validation. Could you explain why you chose not to include any numerical experiments? Even basic examples—successful or not—would help make the paper more accessible to a broader audience. No discussion of robustness or practical performance.**
>
> Response to reviewer: Thanks for raising this question.  We initially focused on establishing theoretical guarantees and approximation bounds for the $k$-means of lines problem based on the local search algorithm. Moreover, we fully agree that numerical experiments can greatly enhance the accessibility and practical relevance of our results. In response to this suggestion, we include a set of numerical experiments as follows.
>
> Experimental Setup: We give comparisons between our local search algorithm and the coreset-based method from [1] on both synthetic datasets (SYN1 with $n$=5000, $d$=10, and SYN2 with $n$=10000, $d$=5) and a real-world Open Street Map dataset used in [1].
> For the Open Street Map dataset, we follow [1] and obtain two road dataset: Region 1 ($n$=476, $d$=2) and Region 2 ($n$=418, $d$=2).
> For the coreset-based method, we compress the data with their coreset algorithm and select $k$ centers via sampling or exhaustive search.
> For our algorithm in $\mathbb{R}^d$, we design a sampling strategy that selects a subset of points from CrossLine to improve computational efficiency.
> The number of samples is set to 100. For each dataset, we run both algorithms 10 times and report minimum cost, maximum cost, and average cost, the standard deviation, and the runtime.
>
> The results show that our algorithms always achieve the best performance compared with coreset+sampling algorithm and coreset+exhaustive algorithm.  In particular, on datasets containing more than 5000 input lines, our algorithm runs at least 43 times faster while maintaining or improving clustering quality. Furthermore, the consistently low standard deviation observed across all experiments demonstrates the robustness of our algorithm.
>
> Table 1: Experimental results of our algorithm and the coreset-based method.
>
> |Dataset|Method|$k$|Min_cost|Max_cost|Mean_cost|Std|time|
> |-|-|-|-|-|-|-|-|
> |RE1|Ours|10|**8.37E-10**|**1.72E-06**|**1.01E-07**|**2.73E-07**|**1.05**|
> |RE1|Coreset+sampling|10|8.17E-04|3.08E-02|6.24E-03|5.89E-03|6.05|
> |RE1|Coreset+exhaustive search|10|-|-|-|-|Over 12 hours|
> |RE1|Ours|3|**5.67E-06**|**1.15E-03**|**1.35E-04**|**2.00E-04**|**0.71**|
> |RE1|Coreset+sampling |3|1.57E-02|8.78E-01|1.39E-01|1.95E-01|4.08|
> |RE1|Coreset+exhaustive search |3|1.10E-03|2.11E-03|1.56E-03|3.01E-04|93.97|
> |RE2|Ours|10|**1.35E-08**|**3.16E-05**|**1.60E-06**|**4.66E-06**|**1.40**|
> |RE2|Coreset+sampling|10|2.89E-03|1.20E-01|2.16E-02|2.13E-02|3.34|
> |RE2|Coreset+exhaustive search |10|-|-|-|-|Over 12 hours|
> |RE2|Ours|3|**1.45E-05**|**5.59E-03**|**1.39E-03**|**1.18E-03**|**0.43**|
> |RE2|Coreset+sampling|3|4.88E-02|6.11E+00|6.21E-01|1.08E+00|4.91|
> |RE2|Coreset+exhaustive search|3|4.69E-04|4.83E-03|2.51E-03|1.09E-03|2.59E+02|
> |SYN1|Ours|10|**1.84E+04**|**1.88E+04**|**1.86E+04**|**1.97E+02**|**4.56E+02**|
> |SYN1|Coreset+sampling|10|3.97E+04|4.51E+04|4.24E+04|2.70E+03|2.04E+04|
> |SYN1|Coreset+exhaustive search|10|-|-|-|-|Over12 hours|
> |SYN2|Ours|3|**1.87E+04**|**1.98E+04**|**1.98E+04**|**5.45E+02**|**1.15E+03**|
> |SYN2|Coreset+sampling|3|-|-|-|-|Over 12 hours|
> |SYN2|Coreset+exhaustive search|3|-|-|-|-|Over 12 hours|
>
>
>
> [1] Marom Y, Feldman D. $k$-Means clustering of lines for big data. NeurIPS 2019.

---

### Official Review · Reviewer_n7xV · 2025-07-02

**Clarity:** 2
**Significance:** 3
**Originality:** 3
**Rating:** 5
**Confidence:** 4

**Summary:**

This paper addresses the $k$-means clustering problem, where the input objects are lines in Euclidean space rather than points. The authors propose a 500-approximation algorithm using a local search framework, which is adapted in a non-trivial way to handle the geometric and analytical challenges posed by clustering lines.

The algorithm proceeds in two main steps:
1. Initialization via coreset: It begins with a coreset-based algorithm from [Marom and Feldman (2019)], originally developed for the $k$-median of lines. Although this coreset was designed for a different objective, the authors show that it yields a constant-factor approximation for the $k$-means of lines problem—though the precise approximation ratio is not specified.
2. Local search: The solution is then improved using a standard local search approach—specifically, center swapping—adapted to the line setting. The candidate centers are constructed by sampling two lines with probability proportional to their current cost and creating a fine grid near their intersection point. This process is repeated a polynomial number of times.

A major analytical challenge is that a single line can be simultaneously close to multiple centers, even if those centers are far apart from each other, making it difficult to directly apply the triangle inequality. Furthermore, since centers can be located anywhere in continuous space, the analysis requires careful discretization.

The high-level argument behind the approximation guarantee is that if the current solution is significantly worse than the optimum (e.g., more than 500× the optimal cost), then there’s a good chance that the weighted-sampled lines will be close to optimal centers. If these lines are near the same optimal center, then their intersection is a promising candidate for placing a new center. This allows the algorithm to iteratively improve the solution by exploring neighborhoods around such intersections.

**Questions:**

### Major comments
- Assumption on non-parallel lines: The paper assumes all input lines are pairwise non-parallel. In practical settings (e.g., road networks, trajectories), parallel lines are common. Is there a simple extension of your algorithm to handle such inputs?
- Radius parameter in CrossLine structure: The definition relies on a radius $r$ tied to the optimal cost, which is unknown in practice. How is $r$ chosen algorithmically? Can it be estimated in a data-dependent or iterative way?
- Candidate center construction: Lemma 8 suggests that optimal centers are near intersections of two nearby lines. Since there are only $O(n^2)$ such intersections, could you simply use all of them as candidate centers for classical point-based local search? Would this simplify the algorithm or analysis?

### Minor comments
- Line 67–68: The description appears to misrepresent paper [9].
- Line 70–72: The summary of paper [10] does not seem accurate.
- Line 200–202: Citation [22] does not pertain to local search.
- Line 211: “near to $c^*$” → “near $c^*$”
- Line 210–213: The claim about the failure of capture relations in the line setting is not entirely clear. A visual counterexample would help clarify this point.
- Line 231–237: Regarding type-2 matched swap pairs: if a center $c$ captures no optimal centers, you pair it with an optimal center “borrowed” from a richer center $c’$. What becomes of $c’$—is it then a type-1 or type-2 pair? This edge case deserves clarification.
- Line 250 (Definition 2): Please clarify the notion of “reassignment cost”—is this just an upper bound on the cost increase when removing a center?
- Lines 110, 270, 274, 292: Typo—“closed to” → “close to”
- Line 292: The inequality $\Delta(L(\psi_S(c)),C) \geq 9\cdot \Delta (L(\psi_S(c)),\{\psi_S(c)\})$ based on Definition 3 is not immediately obvious. Please expand on how Definition 3 implies this bound.

**Ethical Concerns:**

["NO or VERY MINOR ethics concerns only"]

**Final Justification:**

The authors have addressed all my questions. After reading all rebuttal-response histories, I decide to keep my current score.

**Limitations:**

Yes

**Quality:**

3

**Strengths And Weaknesses:**

### Strength
- This is the first constant-factor approximation algorithm for the $k$-means of lines problem with an explicit approximation ratio.
- It successfully adapts the local search technique to this geometric setting, which involves non-trivial modifications and theoretical insights.

### Weakness
- The analysis, especially the main proof, is difficult to follow. Breaking it down into smaller, modular lemmas and including more illustrative diagrams would significantly improve readability.
- Due to the complexity and density of the arguments, I was not able to fully verify all parts of the analysis. While the general structure seems plausible, a more accessible presentation would increase confidence in the result.

---

> ### Author Rebuttal · Authors · 2025-07-31
>
> **Major comments：**
>
> **Question 1.	Assumption on non-parallel lines: The paper assumes all input lines are pairwise non-parallel. In practical settings (e.g., road networks, trajectories), parallel lines are common. Is there a simple extension of your algorithm to handle such inputs?**
>
> Response to reviewer: Thanks for raising this question. In our paper, the assumption that all input lines are non-parallel is made primarily for analytical convenience. Indeed, our algorithm naturally extends to inputs that include parallel lines, and the quality of the solution remains largely unaffected. There are two cases to consider: (1) When all input lines are mutually parallel, each line can be projected onto an axis orthogonal to their common direction, thereby reducing the original problem to a 1-median problem over points in one-dimensional space. For this case, we simply check whether all lines are parallel and, if so, solve the resulting 1-median problem on $\mathbb{R}$.  (2) If the input contains at least one non-parallel lines pair, all optimal cluster centers can be selected from the set of pairwise intersections of the input lines in $\mathbb{R}^2$ [1], or near the candidate point set returned by Algorithm 2 in higher dimensions [2]. Therefore, we simply skip parallel line pairs in Step 4 of Algorithm 2 and Step 4 of Algorithm 1.
>
> **Question 2.	Radius parameter in CrossLine structure: The definition relies on a radius r tied to the optimal cost, which is unknown in practice. How is chosen algorithmically? Can it be estimated in a data-dependent or iterative way?**
>
> Response to reviewer: We thank the reviewer for pointing this out. Since the radius $r$ in the CrossLine structure is defined in terms of the (unknown) optimal cost for theoretical analysis, we choose $r$ in practice using a data-driven iterative heuristic. This choice is based on the observation that if $r$ is too small, the optimal center may lie outside the constructed CrossLine, thereby failing to ensure that at least one point is sufficiently close to the optimal center. Conversely, an excessively large radius may cause all points returned by the CrossLine structure to lie far from the optimal center, resulting in failed single-swap steps. Motivated by Lemma 8, which states that the optimal center lies near the intersection of two nearby lines, we iteratively expand the radius around the intersection. Specifically, we start from a small initial radius (e.g., 0.1 times the minimum distance between the intersection to the point set returned by CrossLine) and iteratively increase it by a fixed step size until the swap succeeds or the radius reaches a predefined upper bound (e.g., the minimum distance between the intersection to the point set returned by CrossLine).
>
>
> **Question 3.	Candidate center construction: Lemma 8 suggests that optimal centers are near intersections of two nearby lines. Since there are only such intersections, could you simply use all of them as candidate centers for classical point-based local search? Would this simplify the algorithm or analysis?**
>
> Response to reviewer: We thank the reviewer for this thoughtful question. While these intersections are geometrically meaningful, we cannot simply use all of them as candidate centers for classical point-based local search, since the algorithm optimizes point-to-point distances rather than point-to-line distances, which are fundamentally different in the line clustering setting. Specifically, if classical point-based local search is directly applied to all intersection points, the optimization focuses on minimizing distances between intersections and centers, rather than between lines and centers. Since a line may yield multiple intersections with other lines, point-based local search may assign these intersections to different centers, leading to ambiguity in determining which center the line should be associated with. Therefore, applying all intersections to a classical point-based local search algorithm does not simplify the algorithm or the analysis.
>
>
> **Minor comments：**
>
> **Question 1.	Line 67–68: The description appears to misrepresent paper [9].**
>
> Response：We appreciate the reviewer for raising this point. We apologize for this misrepresentation and will correct the statement in the revised version to accurately reflect the contributions of [9].
>
> **Question 2.	Line 70–72: The summary of paper [10] does not seem accurate.**
>
> Response：Thank you for pointing this out. We apologize for the inaccurate summary and will correct the description in the revised version to more accurately reflect the contributions of [10].
>
> **Question 3.	Line 200–202: Citation [22] does not pertain to local search.**
>
> Response: We thank the reviewer for pointing this out. Upon careful verification, we agree that citation [22] was mistakenly cited in this context and does not relate to local search. The correct reference should be [18], which provides the relevant analysis of the local search algorithm for classical point-based clustering. We will revise the manuscript accordingly to ensure accurate attribution.
>
>
> **Question 4. Line 211: "near to $c^*$ → "near $c^*$"**
>
> Response: Thank you for pointing out the grammatical errors. We will correct these issues in the revised version and perform a thorough proofreading of the full paper.
>
> **Question 5.	Line 210–213: The claim about the failure of capture relations in the line setting is not entirely clear. A visual counterexample would help clarify this point.**
>
> Response: Thank you for the suggestion. We agree that the explanation was not sufficiently clear. In the revised version, we will include a simple visual counterexample to illustrate why the capture relation may fail in the line setting due to directionality and angular differences.
>
> **Question 6.	Line 231–237: Regarding type-2 matched swap pairs: if a center captures no optimal centers, you pair it with an optimal center “borrowed” from a richer center c’. What becomes of—is it then a type-1 or type-2 pair? This edge case deserves clarification.**
>
> Response: Thanks for raising this question. We state that this case still corresponds to a type-2 matched pair, since according to the definition of proportional capture relation, the borrowed optimal center is not captured by the center $c'$, i.e., the  borrowed optimal center does not receives the largest fraction of lines from that center. We will clarify this point in the revised version.
>
> **Question 7.	Line 250 (Definition 2): Please clarify the notion of “reassignment cost”—is this just an upper bound on the cost increase when removing a center?**
>
> Response: Thanks for raising this question. We state that the "reassignment cost" of Definition 2 is simply an upper bound on the increase in clustering cost when a center is removed, and we will clarify this explicitly in the revised version.
>
> **Question 8.	Lines 110, 270, 274, 292: Typo "closed to" → "close to"**
>
> Response: We appreciate the reviewer for pointing out these typographical errors. We will correct these typos in the revised manuscript.
>
> **Question 9.	Line 292: The inequality based on Definition 3 is not immediately obvious. Please expand on how Definition 3 implies this bound.**
>
> Response: We thank the reviewer for pointing this out. We acknowledge that there was a notation error in Definitions 3 and 5 regarding the characterization of good type-1 and type-2 clusters. These definitions are intended to be consistent with those in [3], and we will correct the presentation accordingly in the revised version.
> Formally, we state that for any $c \in C_S$ , the cluster $L(c)$ is a good type-1 cluster if
>
> $\Delta(L(\psi_S(c)),C)-\eta_1(c)-9\Delta(L(\psi_S(c)),\{\psi_S(c)\})>\frac{1}{100k}\Delta(L,C)$.
>
> A good type-2 cluster is defined analogously. Based on the corrected definitions, we observe that the reassignment cost $\eta_1(c)$ is non-negative, and thus the inequality in Line 292 follows directly.
>
> [1] Tomer Perets. Clustering of lines. Open University of Israel, 2011.
>
> [2] Marom Y, Feldman D. $k$-Means clustering of lines for big data. NeurIPS 2019.
>
> [3] Lattanzi S, Sohler C. A better $k$-means++ algorithm via local search. ICML 2019.

---

> > ### Comment · Reviewer_n7xV · 2025-08-04
> >
> > Thanks for the response. I want to clarify my question 3
> >
> > > Candidate center construction: Lemma 8 suggests that optimal centers are near intersections of two nearby lines. Since there are only $O(n^2)$ such intersections, could you simply use all of them as candidate centers for classical point-based local search? Would this simplify the algorithm or analysis?
> >
> > I'm not saying that you do a point-based local search to cluster centers and intersection points. What I mean is: instead of "*sampling two lines with some probability proportional and creating a fine grid near their intersection point*" to construct candidate centers, we directly list all possible candidate centers around all intersection points (since there are polynomially many).  When doing center-swap we still use distance between centers and lines, just now the candidate center set is fixed rather than sampled on-the-fly.

---

> > > ### Author Response · Authors · 2025-08-05
> > >
> > > **Question: Candidate center construction: Lemma 8 suggests that optimal centers are near intersections of two nearby lines. Since there are only $O(n^2)$ such intersections, could you simply use all of them as candidate centers for classical point-based local search? Would this simplify the algorithm or analysis?**
> > >
> > > **I'm not saying that you do a point-based local search to cluster centers and intersection points. What I mean is: instead of "sampling two lines with some probability proportional and creating a fine grid near their intersection point" to construct candidate centers, we directly list all possible candidate centers around all intersection points (since there are polynomially many). When doing center-swap we still use distance between centers and lines, just now the candidate center set is fixed rather than sampled on-the-fly.**
> > >
> > > **Response**：We thank the reviewer for the insightful suggestion. We acknowledge that enumerating all intersections of two nearby lines as candidate centers can simplify the algorithm in two-dimensional setting, where optimal centers are known to lie exactly at these intersections [1], making the algorithm both computationally efficient and theoretically justified.
> > >
> > > However, in high-dimensional setting, as optimal centers are generally only near, but not exactly located at, intersection points [2], enumerating all intersections of two nearby lines as candidate centers is insufficient to maintain theoretical guarantees. Specifically, in our analysis, Lemma 8 states that with constant probability, the point set returned by CrossLine contains a grid point within distance $r$ from an optimal center. Note that this grid point is not necessarily an intersection point. This property is essential for proving Lemma 3, which ensures that each single-swap operation reduces the cost by at least a $1/(100k)$ fraction. In contrast, when relying solely on enumerated intersection points, we cannot guarantee that any intersection point lies within distance $r$ from the optimal center. Indeed, in the worst case, the nearest enumerated intersection point to the optimal center may be as far as $2r$ away (i.e., the maximum distance from the intersection point to a grid point returned by the CrossLine).
> > > In such cases, Lemma 8 no longer holds, and the reduction cost of a single-swap operation can no longer be bounded, thereby undermining the approximation guarantees of the algorithm.
> > >
> > > Therefore, the sampling procedure and CrossLine construction used in our algorithm are essential to ensure both approximation quality and theoretical guarantees in high-dimensional setting.
> > >
> > > [1] Tomer Perets. Clustering of lines. Open University of Israel, 2011.
> > >
> > > [2] Marom Y, Feldman D.  $k$-Means clustering of lines for big data. NeurIPS 2019.

---

### Official Review · Reviewer_XTPN · 2025-07-02

**Clarity:** 4
**Significance:** 2
**Originality:** 4
**Rating:** 5
**Confidence:** 2

**Summary:**

This paper studies the $k$-means of lines problem. Compared to the original $k$-means problem, the dataset $P$ consists of $n$ lines rather than points. The algorithm needs to find $k$ centers $C \subset \mathbb{R}^d$ to minimize $\sum_{p\in P}\min_{c\in C}d(p,c)^2$, where $d(p,c)$ denotes the distance between line $p$ and point $c$.

This paper proposes the first local search algorithm (with a theoretical guarantee) for the $k$-means problem. Local search is widely used for designing efficient clustering algorithms. However, it is hard to generalize the local search algorithms for the original $k$-means problem to the line setting, since the geometric properties of lines are very different from points.

In order to overcome the challenges, the authors introduce a new mapping between the current solution and the optimal solution.
They also designed a new sampling procedure which discretizes the solution space into a finite set of representative points so that they can sample candidate centers efficiently.

**Questions:**

See weakness.

**Ethical Concerns:**

["NO or VERY MINOR ethics concerns only"]

**Final Justification:**

The authors explain the hardness of the problem, which answers my question in weakness.

**Limitations:**

yes

**Paper Formatting Concerns:**

No major foramtting issues.

**Quality:**

4

**Strengths And Weaknesses:**

Strength:

1. This paper proposes the first constant-approximate algorithm for the $k$-means problem in polynomial time. In order to apply local search to the $k$-means of lines problem, the authors develop several new techniques to overcome the complicated geometric properties of lines. These techniques may also be used in the following studies of $k$-means of lines or even other geometric problems.

2. Their algorithm is quite simple and efficient in practice (in low-dimensional cases).

Weakness:

The running time of the algorithm has an exponential dependency on $d$, which means the algorithm might not be polynomial time in high-dimensional cases.

---

> ### Author Rebuttal · Authors · 2025-07-31
>
> **Question：The running time of the algorithm has an exponential dependency on $d$, which means the algorithm might not be polynomial time in high-dimensional cases.**
>
> Response: We apologize for the confusion. For clustering and related problems, the running time are always related to the input size $n$, dimension $d$, and cluster numbers $k$. Especially, for many PTAS approximation algorithms, which are called polynomial time approximation, the parameters $d$ and $k$ appears in exponential form in running time. For example, $O(n(\log n)^k\epsilon^{-2k^2d})$ for $k$-means [1], $O(k\epsilon^{-d}\log^{2d+2}n)$ for $k$-means and $k$-median [2]. The reason that all   those algorithms are called polynomial time approximation is based on the fact that the parameters $d$ and $k$ are assumed to be fixed constants.
>
> For the line-clustering problems, discretization and geometric covering methods are usually used to design algorithms with theoretical guarantees, which always results in the exponential dependence on parameter dimension $d$ and the number of centers $k$. Following the routine of designing PTAS for clustering problem, in our paper, we view the parameters $d$ and $k$ as fixed constants. Although we have $9^d$ in the running time, it is still polynomial time on input size $n$.
>
> Moreover, we conduct experiments to illustrate our algorithm performance (see Table 1). It can be seen that our proposed local search algorithm achieves much better performance with a large reduction in running time.   In particular, on datasets containing more than 5000 input lines, our algorithm runs at least 43 times faster while maintaining or improving clustering quality.
>
> Experimental Setup: We give comparisons between our local search algorithm and the coreset-based method from [3] on both synthetic datasets (SYN1 with $n$=5000, $d$=10, and SYN2 with $n$=10000, $d$=5) and a real-world Open Street Map dataset used in [3].
> For the Open Street Map dataset, we follow [3] and obtain two road dataset: Region 1 ($n$=476, $d$=2) and Region 2 ($n$=418, $d$=2).
> For the coreset-based method, we compress the data with their coreset algorithm and select $k$ centers via sampling or exhaustive search.
> For our algorithm in $\mathbb{R}^d$, we design a sampling strategy that selects a subset of points from CrossLine to  improve computational efficiency.
> The number of samples is set to 100. For each dataset, we run both algorithms 10 times and report minimum cost, maximum cost, and average cost, the standard deviation, and the runtime.
>
> Table 1: Experimental results of our algorithm and the coreset-based method.
> |Dataset|Method|$k$|Min_cost|Max_cost|Mean_cost|Std|time|
> |-|-|-|-|-|-|-|-|
> |RE1|Ours|10|**8.37E-10**|**1.72E-06**|**1.01E-07**|**2.73E-07**|**1.05**|
> |RE1|Coreset+sampling|10|8.17E-04|3.08E-02|6.24E-03|5.89E-03|6.05|
> |RE1|Coreset+exhaustive search|10|-|-|-|-|Over 12 hours|
> |RE1|Ours|3|**5.67E-06**|**1.15E-03**|**1.35E-04**|**2.00E-04**|**0.71**|
> |RE1|Coreset+sampling |3|1.57E-02|8.78E-01|1.39E-01|1.95E-01|4.08|
> |RE1|Coreset+exhaustive search |3|1.10E-03|2.11E-03|1.56E-03|3.01E-04|93.97|
> |RE2|Ours|10|**1.35E-08**|**3.16E-05**|**1.60E-06**|**4.66E-06**|**1.40**|
> |RE2|Coreset+sampling|10|2.89E-03|1.20E-01|2.16E-02|2.13E-02|3.34|
> |RE2|Coreset+exhaustive search |10|-|-|-|-|Over 12 hours|
> |RE2|Ours|3|**1.45E-05**|**5.59E-03**|**1.39E-03**|**1.18E-03**|**0.43**|
> |RE2|Coreset+sampling|3|4.88E-02|6.11E+00|6.21E-01|1.08E+00|4.91|
> |RE2|Coreset+exhaustive search|3|4.69E-04|4.83E-03|2.51E-03|1.09E-03|2.59E+02|
> |SYN1|Ours|10|**1.84E+04**|**1.88E+04**|**1.86E+04**|**1.97E+02**|**4.56E+02**|
> |SYN1|Coreset+sampling|10|3.97E+04|4.51E+04|4.24E+04|2.70E+03|2.04E+04|
> |SYN1|Coreset+exhaustive search|10|-|-|-|-|Over12 hours|
> |SYN2|Ours|3|**1.87E+04**|**1.98E+04**|**1.98E+04**|**5.45E+02**|**1.15E+03**|
> |SYN2|Coreset+sampling|3|-|-|-|-|Over 12 hours|
> |SYN2|Coreset+exhaustive search|3|-|-|-|-|Over 12 hours|
>
>
>
>
>
>
>
> [1] Jirı Matousek. On approximate geometric $k$-clustering. Discrete & Computational Geometry, 2000.
>
> [2] Har-Peled S, Mazumdar S. On coresets for $k$-means and $k$-median clustering. STOC 2004.
>
> [3] Marom Y, Feldman D. $k$-Means clustering of lines for big data. NeurIPS 2019.

---

### Official Review · Reviewer_jhQS · 2025-07-03

**Clarity:** 3
**Significance:** 2
**Originality:** 2
**Rating:** 3
**Confidence:** 3

**Summary:**

The paper addresses the problem of clustering a set of lines using points as cluster centers and the k-means objective. The idea is to develop an approximation algorithm in the framework of local swaps. The main difficulty is that the point-to-line distance, used in the paper, does not satisfy the triangle inequality, and thus standard local-swap arguments are not amenable to this setting. Another question to address for the method to be operational is the initialization step, for initializing cluster centers, before the local-swap phase. The paper proposes a new algorithm that addresses those questions and demonstrates that the proposed algorithm achieves a constant factor approximation in this local-swap setting. The paper is of mostly theoretical nature and there is no experimental evaluation of the proposed method.

**Questions:**

I welcome answers to weak points W1 and W2 mentioned above.

**Ethical Concerns:**

["NO or VERY MINOR ethics concerns only"]

**Final Justification:**

The paper addresses an intellectually interesting problem and the technical novelty is good, however, I still see a number of weaknesses. In particular, (i) the problem motivation is weak; (ii) the theoretical running time of the method is large; (iii) the theoretical approximation guarantee is large; and (iv) the theoretical improvement is only for a particular family of methods (local search) and it is not general. Another weakness is that the paper originally did not contain any experimental evaluation. The authors provided some experimental results in the rebuttal. My concern is whether providing (the complete set of) experiments in the rebuttal phase goes beyond the scope of what a rebuttal is meant to be. Additionally, there is the issue of fairness to the authors of other submissions. This could be discussed among the area chair and the other reviewers. Overall, given these weakesses and concerns, I am keeping my original score to "borderline reject".

**Limitations:**

The authors make an honest attempt to list the limitations of their problem setting. In addition to the ones listed by the authors, I would also add the weak points mentioned above.

**Paper Formatting Concerns:**

I see no issues with paper formatting.

**Quality:**

3

**Strengths And Weaknesses:**

The main strengths of the paper are the clarity in the problem definition, the rigorous analysis of the proposed method, and the achievement of obtaining a constant-factor approximation for the problem of clustering lines.
On the other hand, I found a few important weaknesses.
W1. I found the motivation of the problem very weak. The paper states as motivation applications in traffic and motion data analysis. However, such real-world data are never straight (infinite lines). Another element missing motivation is the choice of using points as cluster centers. Why not using lines as cluster centers? This looks a more straightforward generalization of the k-means setting for points, where both the data and the cluster centers are points. Overall, I am totally on board that the problem is intellectually interesting and a meaningful generalization of the k-means problem for points, however, I think that for a AI/ML conference like NeurIPS a stronger motivation is needed. Perhaps the paper is a better fit for a venue in theoretical computer science, or computational geometry.
W2. Earlier work provides a PTAS for the same problem, while this paper provides a constant-factor approximation, with a very large constant. Thus, this is not the theoretically best result available. Indeed, the claim of the paper is that this is the first constant-factor approximation for the local-swaps setting, not for the problem itself. It is also true that the PTAS algorithm has high complexity. However, to strengthen the value of the paper as a practical alternative with manageable complexity I would expect to see some experimental evaluation of the proposed method. Additionally, a convincing experimental evaluation would also address the issue of motivation in real-world applications, by forcing the authors to identify meaningful datasets and applications to use the lines-clustering problem and methods. Overall, I appreciate the contribution of "first constant-factor approximation within a specific solution framework" and I value the technical depth and rigorous analysis of the paper, however, in addition to the weak motivation, and lack of experimental evaluation, I think that the overall contribution is not sufficient for NeurIPS.

---

> ### Author Rebuttal · Authors · 2025-07-31
>
> **W1. I found the motivation of the problem very weak. The paper states as motivation applications in traffic and motion data analysis. However, such real-world data are never straight (infinite lines). Another element missing motivation is the choice of using points as cluster centers. Why not using lines as cluster centers? This looks a more straightforward generalization of the k-means setting for points, where both the data and the cluster centers are points. Overall, I am totally on board that the problem is intellectually interesting and a meaningful generalization of the k-means problem for points, however, I think that for a AI/ML conference like NeurIPS a stronger motivation is needed. Perhaps the paper is a better fit for a venue in theoretical computer science, or computational geometry.**
>
> Response: We appreciate the reviewer’s insightful question. Clustering infinite lines with points can be naturally applied to the following scenarios in motion data analysis [1]: in multi-camera systems that observe 3D dynamic objects in complex scenes, each camera provides a directional observation toward an object (i.e., a ray extending from the camera to the observed object.) By representing each such observation as an infinite line in space, a set of lines corresponding to different views can be obtained. Obviously, clustering these infinite lines helps identify different objects and recover object locations, as infinite lines observing the same object tend to intersect at or near the object’s location. Thus, the motivation of clustering infinite lines with points centers is to estimate the locations of multiple latent objects, based on directional information collected from spatially different viewpoints.
>
> Although real-world data are not truly straight or infinite lines, for the multi-camera systems observing 3D dynamic objects, any directional measurement from a fixed viewpoint to an object can be naturally modeled as an infinite line in space. Under those applications, it is reasonable to model the observations as infinite lines. Obviously, for the observation object applications, all the views can be modeled as infinite lines.
>
> For the multi-camera systems observing objects application, since infinite lines observing the same object tend to intersect at or near the object’s location, it is natural to use point as centers to clustering infinite lines. Moreover, since the infinite lines have unbounded nature, if an infinite line is chosen as center, inferring precise object locations in Euclidean space seems not workable.
>
> In the revised version of the paper, we will clearly present the motivation by emphasizing its practical relevance.
>
> **W2. Added experimental evaluation.**
>
> Response: We thank the reviewer for pointing this out. For completeness, we give comparisons between our local search algorithm and the coreset-based method from [2] on both synthetic datasets and a real-world Open Street Map dataset used in [2].
>
> We give comparisons between our local search algorithm and the coreset-based method from [2] on both synthetic datasets (SYN1 with $n$=5000, $d$=10, and SYN2 with $n$=10000, $d$=5) and a real-world Open Street Map dataset used in [2].
> For the Open Street Map dataset, we follow [2] and obtain two road dataset: RE1 ($n$=476, $d$=2) and RE2 ($n$=418, $d$=2).
> For the coreset-based method, we compress the data with their coreset algorithm and select $k$ centers via sampling or exhaustive search.
> For our algorithm in $\mathbb{R}^d$, we design a sampling strategy that selects a subset of points from CrossLine to improve computational efficiency.
> The number of samples is set to 100. For each dataset, we run both algorithms 10 times and report minimum cost, maximum cost, and average cost, the standard deviation, and the runtime.
>
> The results show that our algorithms always achieve the best performance compared with coreset+sampling algorithm and coreset+exhaustive algorithm. In particular, on datasets containing more than 5000 input lines, our algorithm runs at least 43 times faster while maintaining or improving clustering quality.
>
> Table 1: Experimental results of our algorithm and the coreset-based method.
>
> |Dataset|Method|$k$|Min_cost|Max_cost|Mean_cost|Std|time|
> |-|-|-|-|-|-|-|-|
> |RE1|Ours|10|**8.37E-10**|**1.72E-06**|**1.01E-07**|**2.73E-07**|**1.05**|
> |RE1|Coreset+sampling|10|8.17E-04|3.08E-02|6.24E-03|5.89E-03|6.05|
> |RE1|Coreset+exhaustive search|10|-|-|-|-|Over 12 hours|
> |RE1|Ours|3|**5.67E-06**|**1.15E-03**|**1.35E-04**|**2.00E-04**|**0.71**|
> |RE1|Coreset+sampling |3|1.57E-02|8.78E-01|1.39E-01|1.95E-01|4.08|
> |RE1|Coreset+exhaustive search |3|1.10E-03|2.11E-03|1.56E-03|3.01E-04|93.97|
> |RE2|Ours|10|**1.35E-08**|**3.16E-05**|**1.60E-06**|**4.66E-06**|**1.40**|
> |RE2|Coreset+sampling|10|2.89E-03|1.20E-01|2.16E-02|2.13E-02|3.34|
> |RE2|Coreset+exhaustive search |10|-|-|-|-|Over 12 hours|
> |RE2|Ours|3|**1.45E-05**|**5.59E-03**|**1.39E-03**|**1.18E-03**|**0.43**|
> |RE2|Coreset+sampling|3|4.88E-02|6.11E+00|6.21E-01|1.08E+00|4.91|
> |RE2|Coreset+exhaustive search|3|4.69E-04|4.83E-03|2.51E-03|1.09E-03|2.59E+02|
> |SYN1|Ours|10|**1.84E+04**|**1.88E+04**|**1.86E+04**|**1.97E+02**|**4.56E+02**|
> |SYN1|Coreset+sampling|10|3.97E+04|4.51E+04|4.24E+04|2.70E+03|2.04E+04|
> |SYN1|Coreset+exhaustive search|10|-|-|-|-|Over12 hours|
> |SYN2|Ours|3|**1.87E+04**|**1.98E+04**|**1.98E+04**|**5.45E+02**|**1.15E+03**|
> |SYN2|Coreset+sampling|3|-|-|-|-|Over 12 hours|
> |SYN2|Coreset+exhaustive search|3|-|-|-|-|Over 12 hours|
>
> [1] Zhang T, Chen X, Wang Y, et al. MUTR3D: A multi-camera tracking framework via 3D-to-2D queries. CVPR 2022.
>
> [2] Marom Y, Feldman D. $k$-Means clustering of lines for big data. NeurIPS 2019.

---

> > ### Comment · Reviewer_jhQS · 2025-08-06
> >
> > Thank you for the response. Regarding the motivating application you are mentioning, I understand that (1) the scenario seem to be better represented by semi-infinite lines, not infinite, as lines of sight do not extend behind the cameras, and (2) in the CVPR 2022 paper you are citing there is no discussion on the problem of clustering of infinite lines; instead they are considering tracking of objects but the techniques are very different.

---

> > > ### Author Response · Authors · 2025-08-07
> > >
> > > **Question: Regarding the motivating application you are mentioning, I understand that (1) the scenario seem to be better represented by semi-infinite lines, not infinite, as lines of sight do not extend behind the cameras, and (2) in the CVPR 2022 paper you are citing there is no discussion on the problem of clustering of infinite lines; instead they are considering tracking of objects but the techniques are very different.**
> > >
> > > Response: We thank the reviewer for pointing out these issues. While the lines of sight from cameras are physically semi-infinite in our mentioned scenario, the observation direction of each line is the key geometric information for clustering, thus modeling the observations as either semi-infinite or infinite lines has no impact on the clustering result. However, using semi-infinite lines (i.e., rays) may introduce inconsistencies in distance computation. According to the standard definition in [1], the point-to-ray distance is the projection distance if the projection of the point lies on the ray; otherwise, it becomes the Euclidean distance to the origin of the ray. In multi-camera scenarios with multiple objects, as an object may lie on the backward extension of a ray not observing it, both types of point-to-ray distance naturally occur, leading to inconsistencies in distance computation. In contrast, modeling observations as infinite lines can ensure consistency in point-to-line distance computation.
> > >
> > > Indeed, camera-based observation is merely one representative scenario of observation-based applications. In broader observation-based applications, such as maritime radar and acoustic sensing, only the observation direction is available, as wave reflections may cause the object to appear in either the forward or backward direction relative to the observation point.
> > >
> > > We apologize for any confusion caused by our citation of the CVPR 2022. Instead of referencing any technique or method for clustering infinite lines, our citation of the CVPR 2022 is intended solely to introduce a real-world scenario from motion data analysis involving multi-camera observation of multiple objects. In this scenario, the resulting observations from different cameras can be naturally modeled as a set of infinite lines in 3D space, as illustrated in Figure 1 of [2]. Therefore, both our problem formulation and solution techniques are naturally different from those used in the tracking task of CVPR 2022.
> > >
> > > [1] Sunday, Dan. Distance between point and line, ray, or line segment. Geometric Tools 1999.
> > >
> > > [2] Marom Y, Feldman D. $k$-Means clustering of lines for big data. NeurIPS 2019.

---

### Official Review · Reviewer_XmFp · 2025-07-03

**Clarity:** 3
**Significance:** 2
**Originality:** 3
**Rating:** 4
**Confidence:** 4

**Summary:**

The paper introduces a local search (LS) algorithm for k-means clustering of lines, where the input are lines in R^d and the k centers are points and the sum of squared distances of the lines to their closest of the k points points are to be minimized. The problem is said to have "received little attention in existing literature".

I find The motivation of the model is vague, and it seems the reverse case has received a lot more attention, where the input consists of (possibly noisy) points from a linear flat and the centers are linear flats trying to reconstruct the true subspaces in which the points lie. Also when considering lines as inputs, why would they extend infinitely instead of just line segments, which would be a special case of polygonal curves clustering?

The problem in extending existing LS algorithms (for k-means) to the lines setting is of technical nature. For instance the tringle inequality does not hold and this complicates tracking the cost changes when reassigning from approximate centers to optimal centers. To this end a novel technique called "proportional capture relation" is introduced that also accounts for the number of reassigned lines to avoid large cost increase.

The second technical problem is in finding candidate swap pairs which is solved by a centroid set construction from previous coreset work [22] and a grid construction around the point where a pair of lines crosses (or attains its smallest distance in high dimensions)

Although everything is claimed to be polynomial time in several places even in the high dimensional setting, I found C^d dependences which are clearly exponential (for the Cross Line grid construction).

Once those technicalities are solved, the analysis follows pretty much verbatim to recent work [18] on local search for the standard k-means of points, and more classic work (I am not sure but the right reference that is missing might be Vijay Arya, Naveen Garg, Rohit Khandekar, Adam Meyerson, Kamesh Munagala, and Vinayaka Pandit. Local search heuristics for k-median and facility
location problems. SIAM Journal on Computing, 33(3):544–562, 2004. )

**Questions:**

- perhaps the authors could comment on the motivation of clustering infinite lines with points, instead of the more established settings of (finite) line segments as in polygonal curve clustering, or clustering points with linear flats as in subspace clustering?

- I would recommend to specify "with polynomial time" in Theorem 1

- I find it confusing to read the \delta(p,q) notation for the **squared** Euclidean distances. why not \delta(p,q)^2 ? Or better |p-q|_2^2!!

- in 3.1 could you explain why is the choice of the smallest distance between two lines good?

- below Lemma 3 two mentions of [22] where it should be [18] ?!

- definnition 6 and running time analysis: 9^d does not look very polynomial to me ?!

**Ethical Concerns:**

["NO or VERY MINOR ethics concerns only"]

**Final Justification:**

The rebuttal and discussion have slightly improved my opinion about the originality and technical strength of this submission. Motivation-wise, the problem seems to have some niche applications and could still be strengthened in the intro. Assumptions and limitations should be discussed more clearly, in particular the high d-dependence and associated limitation to (low) constant dimensions. Additionally, some sort of dimension reduction could strengthen the paper, and the applicability beyond 2D or 3D. I also agree with another reviewer that the paper may be a better fit in the TCS or CG community. Overall, I increased my rating slightly to a "4: Borderline accept".

**Limitations:**

Limitatios section is very limited. I think noise-freeness is minor compared to the *real* limitations such as exponential d dependence, very high approximation factor etc...

**Quality:**

3

**Strengths And Weaknesses:**

*Strength*
- local search helps in the new setting
- simple algorithm

*Weakness*
- weak motivation
- high dimension dependence (C^d)
- bad approximation factor 500 in expectation (so 1500 with probability 2/3 ?)
- analysis very close to existing local search literature
- lots of typos

---

> ### Author Rebuttal · Authors · 2025-07-31
>
> **W1&Q1: regarding weak motivation.**
>
> Response: We appreciate the reviewer’s insightful question. Clustering infinite lines with points has background application in Computer Vision [1-2]. For example, in the scenario using multiple cameras to observe fixed objects, it can be viewed from the objects that each camera has directional observations (i.e., direction from camera to the object), and a set of infinite lines can be obtained by considering each camera with direction to the corresponding object. Obviously, clustering these infinite lines helps identify different objects and recover their locations, as infinite lines observing the same object tend to intersect at or near the object's location.
>
> Thus, the motivation of clustering infinite lines with points is to estimate the locations of multiple latent objects, based on directional information collected from spatially different viewpoints.
>
> In contrast, curve and subspace clustering use curves and linear subspaces as centers, which do not support precise localization of objects. Moreover, infinite lines naturally encode unbounded directional information, which cannot be captured by finite segments, curves, or subspaces used in curve and subspace clustering.
>
> In revised version, we will expand the motivation by emphasizing its practical relevance and clarifying its distinction from existing clustering paradigms.
>
> **W2&Q6: regarding high dimension dependence.**
>
> Response: We apologize for the confusion. For clustering and related problems, the running time are always related to the input size $n$, dimension $d$, and cluster numbers $k$. Especially, for many PTAS approximation algorithms, which are called polynomial time approximation, the parameters $d$ and $k$ appear in exponential form in running time. For example, $O(n(\log n)^k\epsilon^{-2k^2d})$ for $k$-means [3], $O(k\epsilon^{-d}\log^{2d+2}n)$ for $k$-means and $k$-median [4]. The reason that all those algorithms are called polynomial time approximation is based on the fact that the parameters $d$ and $k$ are assumed to be fixed constants.
>
> For line-clustering problems, discretization and geometric covering methods are usually used to design algorithms with theoretical guarantees, which always results in the exponential dependence on parameter dimension $d$ and the number of centers $k$. Following the routine of designing PTAS for clustering problems, in our paper, we view the parameters $d$ and $k$ as fixed constants. Although we have $9^d$ in the running time, it is still polynomial time on input size $n$.
>
> **W3: bad approximation factor 500 in expectation (so 1500 with probability 2/3?)**
>
> Response: Thanks for pointing out this. Local search has been applied extensively to solve clustering and related problems, and many of them get approximation guarantees in expectation (e.g., 509 in [5], over $10^9$ in [6], and over 2000 in [7]). In this paper, we apply local search to solve the $k$-means of lines problem. Following the routine in the literature, we analyze the approximation factor in expectation. We agree with the reviewer that based on Markov's inequality, with probability at least 2/3, the factor is 1500. Intuitively, line-clustering is much harder than $k$-means or $k$-median. However, our expected approximation factor 500 is much better than the results in [6,7].
>
> Although our approximation factor in expectation seems not good enough, we conduct experiments to show  the performance of our algorithm (see Table 1). It can be seen that our local search algorithm achieves much better performance with a large reduction in running time.
>
> Experimental Setup: We give comparisons between our local search algorithm and the coreset-based method from [1] on both synthetic datasets (SYN1 with n=5000 d=10, and SYN2 with n=10000 d=5) and a real-world Open Street Map dataset (RE1 with n=476 d=2, and RE2 with n=418 d=2) used in [1].
> For coreset-based method, we compress the data with their coreset algorithm and select $k$ centers via sampling or exhaustive search.
> For our algorithm, we design a sampling strategy that selects a subset of 100 points from CrossLine to improve computational efficiency.
> For each dataset, we run both algorithms 10 times and report minimum cost, maximum cost, and average cost, the standard deviation, and the runtime.
>
> Table 1: Experimental results of our algorithm and the coreset-based method.
> |Dataset|Method|$k$|Min_cost|Max_cost|Mean_cost|Std|time|
> |-|-|-|-|-|-|-|-|
> |RE1|Ours|10|**8.37E-10**|**1.72E-06**|**1.01E-07**|**2.73E-07**|**1.05**|
> |RE1|Coreset+sampling|10|8.17E-04|3.08E-02|6.24E-03|5.89E-03|6.05|
> |RE1|Coreset+exhaustive search|10|-|-|-|-|Over 12 hours|
> |RE1|Ours|3|**5.67E-06**|**1.15E-03**|**1.35E-04**|**2.00E-04**|**0.71**|
> |RE1|Coreset+sampling |3|1.57E-02|8.78E-01|1.39E-01|1.95E-01|4.08|
> |RE1|Coreset+exhaustive search |3|1.10E-03|2.11E-03|1.56E-03|3.01E-04|93.97|
> |RE2|Ours|10|**1.35E-08**|**3.16E-05**|**1.60E-06**|**4.66E-06**|**1.40**|
> |RE2|Coreset+sampling|10|2.89E-03|1.20E-01|2.16E-02|2.13E-02|3.34|
> |RE2|Coreset+exhaustive search |10|-|-|-|-|Over 12 hours|
> |RE2|Ours|3|**1.45E-05**|**5.59E-03**|**1.39E-03**|**1.18E-03**|**0.43**|
> |RE2|Coreset+sampling|3|4.88E-02|6.11E+00|6.21E-01|1.08E+00|4.91|
> |RE2|Coreset+exhaustive search|3|4.69E-04|4.83E-03|2.51E-03|1.09E-03|2.59E+02|
> |SYN1|Ours|10|**1.84E+04**|**1.88E+04**|**1.86E+04**|**1.97E+02**|**4.56E+02**|
> |SYN1|Coreset+sampling|10|3.97E+04|4.51E+04|4.24E+04|2.70E+03|2.04E+04|
> |SYN1|Coreset+exhaustive search|10|-|-|-|-|Over12 hours|
> |SYN2|Ours|3|**1.87E+04**|**1.98E+04**|**1.98E+04**|**5.45E+02**|**1.15E+03**|
> |SYN2|Coreset+sampling|3|-|-|-|-|Over 12 hours|
> |SYN2|Coreset+exhaustive search|3|-|-|-|-|Over 12 hours|
>
> **W4: analysis very close to existing local search literature.**
>
> Response: We thank the reviewer for the comment. The local search has been applied to solve many clustering related problem, such as $k$-means [5-7], $k$-means with outlier [8-9], fair-range clustering [10]) or its related problems (e.g., subset selection [11]). It can be seen that the algorithm of applying local search always contains several routine steps, and seems similar for different problems. The major obstacle of local search is how to analyze the relation between local optimal solution and optimal solution, how to construct the swap pairs, and how to analyze the success probability, etc. It can be seen from [8-11] that for different problems, the theoretical guarantee analysis challenge may be totally different.
>
> For line-clustering problem, the analysis of our local search process contains the following obstacles: (1) triangle inequality cannot be applied for point-to-line distance, which poses a big challenge for local search analysis; (2) it is difficult to identify effective swap points in infinite geometric space.
>
> Because of the above obstacles, it is a non-trivial task to apply local search to solve the line-clustering problem. As far as we know, there is no available local search-based results in literature. To overcome these obstacles, in this paper, we propose a new method, called proportional capture relation, to bypass the triangle inequality barrier, and present a new structure, called CrossLine, to ensure the coverage of high-quality swap points for local search.
>
> **Q2: I would recommend to specify "with polynomial time" in Theorem 1.**
>
> Response: We thank the reviewer for the helpful suggestion. In revised version, we will explicitly state that the algorithm runs in polynomial time with respect to the input size $n$, under the assumption that the dimension $d$ is treated as a fixed constant.
>
> **Q3: regarding the squared Euclidean distances.**
>
> Response：Thanks for pointing this out. In revised version, we will replace the current notation with the more standard form $|p-q|_2^2$ to enhance clarity and consistency.
>
> **Q4: in 3.1 could you explain why is the choice of the smallest distance between two lines good?**
>
> Response：Thanks for raising this question. In Section 3.1, we select the pair of lines with the smallest mutual distances, as this leads to an initial solution with an approximation guarantee. This conclusion is based on a key property given in Lemma 16 of [1]: Let $P$ be the candidate point set (i.e., the point set from the pair of lines with the smallest mutual distance). Then, selecting any $k$ points from $P$ suffices to approximate the optimal solution within a large constant factor. In local search, an initial solution with an approximation guarantee is critical, as the lack of such a guarantee can lead to excessive iterations and render the overall optimization process difficult to analyze.
>
> **Q5: below Lemma 3 two mentions of [22] where it should be [18]?!**
>
> Response: We agree with the reviewer that both citations of [22] below Lemma 3 are incorrect, and should have referred to [18]. We will correct all those in revised version.
>
> [1]Marom Y, Feldman D. $k$-Means clustering of lines for big data. NeurIPS 2019.
>
> [2]Lotan S, Sanches Shayda E E, Feldman D. Coreset for line-sets clustering. NeurIPS 2022.
>
> [3]Matousek J. On approximate geometric $k$-clustering. Discrete & Computational Geometry 2000.
>
> [4]Har-Peled S, Mazumdar S. On coresets for $k$-means and $k$-median clustering. STOC 2004.
>
> [5]Lattanzi S, Sohler C. A better $k$-means++ algorithm via local search. ICML 2019.
>
> [6]Choo D, Grunau C, Portmann J, et al. $k$-means++: few more steps yield constant approximation. ICML 2020.
>
> [7]Fan C, Li P, Li X. LSDS++: Dual sampling for accelerated $k$-means++. ICML 2023.
>
> [8]Huang J, Feng Q, Huang Z, et al. Near-linear time approximation algorithms for $k$-means with outliers. ICML 2024.
>
> [9]Grunau C, Rozhon V. Adapting $k$-means algorithms for outliers. ICML 2022.
>
> [10]Zhang Z, Chen X, Liu L, et al. Parameterized approximation schemes for fair-range clustering. NeurIPS 2024.
>
> [11]Zou Y, Huang Z, Xu J, et al. Linear time approximation algorithm for column subset selection with local search. NeurIPS 2024.

---

> > ### Comment · Reviewer_XmFp · 2025-08-06
> >
> > Thank you very much for your reply and clarification of my questions. I consider it extremely important to clarify assumptions such as fixed (and low) parameters $d,k$ and the exact meaning of “polynomial time” in the context of your work, and to discuss the limitations regarding these parameters and the large approximation ratio, rather than noise-freeness, which seems “artificial” and unimportant given the above mentioned serious limitations of the method and analysis. However, I admit that the analysis is somewhat more demanding than I had originally assumed, and I will reconsider my assessment in the final review phase.

---

> > > ### Author Response · Authors · 2025-08-07
> > >
> > > We sincerely thank the reviewer for the thoughtful and detailed feedback, as well as the constructive suggestions regarding the assumptions and limitations of our work. In the revised version, we will clarify all relevant assumptions and explicitly discuss the limitations of our work.

---

### Decision · Program_Chairs · 2025-09-17

**Decision:**

Accept (poster)

**Comment:**

The authors study the k-means on line problem that is a natural extension of the classic k-means problem and captures many natural scenarios.

In particular, they propose a local search algorithm to obtain a poly time approximation algorithm for the problem.

The committee found the problem and the results interesting, although few  improvements should be implemented before publication:
- the paper contains multiple typos that should be fixed.
- the experiments should be included in the final version of the paper.
- a longer discussion on the practical motivation of the paper should be added to the paper.